# On the Convergence of FedAvg on Non-IID Data

**Xiang Li**[*]
School of Mathematical Sciences
Peking University
Beijing, 100871, China
`smslixiang@pku.edu.cn`

**Kaixuan Huang**[*]
School of Mathematical Sciences
Peking University
Beijing, 100871, China
`hackyhuang@pku.edu.cn`

**Wenhao Yang**[*]
Center for Data Science
Peking University
Beijing, 100871, China
`yangwenhaosms@pku.edu.cn`

**Shusen Wang**
Department of Computer Science
Stevens Institute of Technology
Hoboken, NJ 07030, USA
`shusen.wang@stevens.edu`

**Zhihua Zhang**
School of Mathematical Sciences
Peking University
Beijing, 100871, China
`zhzhang@math.pku.edu.cn`

## Abstract

Federated learning enables a large amount of edge computing devices to jointly learn a model without data sharing. As a leading algorithm in this setting, Federated Averaging (`FedAvg`) runs Stochastic Gradient Descent (SGD) in parallel on a small subset of the total devices and averages the sequences only once in a while. Despite its simplicity, it lacks theoretical guarantees under realistic settings. In this paper, we analyze the convergence of `FedAvg` on non-iid data and establish a convergence rate of $\mathcal{O}(\frac{1}{T})$ for strongly convex and smooth problems, where $T$ is the number of SGDs. Importantly, our bound demonstrates a trade-off between communication-efficiency and convergence rate. As user devices may be disconnected from the server, we relax the assumption of full device participation to partial device participation and study different averaging schemes; low device participation rate can be achieved without severely slowing down the learning. Our results indicates that heterogeneity of data slows down the convergence, which matches empirical observations. Furthermore, we provide a necessary condition for `FedAvg` on non-iid data: the learning rate $\eta$ must decay, even if full-gradient is used; otherwise, the solution will be $\Omega(\eta)$ away from the optimal.

## 1 Introduction

Federated Learning (FL), also known as federated optimization, allows multiple parties to collaboratively train a model without data sharing (Konevcnỳ et al., 2015; Shokri and Shmatikov, 2015; McMahan et al., 2017; Konevcnỳ, 2017; Sahu et al., 2018; Zhuo et al., 2019). Similar to the centralized parallel optimization (Jakovetic, 2013; Li et al., 2014a;b; Shamir et al., 2014; Zhang and Lin, 2015; Meng et al., 2016; Reddi et al., 2016; Richtárik and Takác, 2016; Smith et al., 2016; Zheng et al., 2016; Shusen Wang et al., 2018), FL let the user devices (aka worker nodes) perform most of the computation and a central parameter server update the model parameters using the descending directions returned by the user devices. Nevertheless, FL has three unique characters that distinguish it from the standard parallel optimization Li et al. (2019).

---

[*]Equal contribution.

First, the training data are massively distributed over an incredibly large number of devices, and the connection between the central server and a device is slow. A direct consequence is the slow communication, which motivated communication-efficient FL algorithms (McMahan et al., 2017; Smith et al., 2017; Sahu et al., 2018; Sattler et al., 2019). Federated averaging (FedAvg) is the first and perhaps the most widely used FL algorithm. It runs $E$ steps of SGD in parallel on a small sampled subset of devices and then averages the resulting model updates via a central server once in a while.[1] In comparison with SGD and its variants, FedAvg performs more local computation and less communication.

Second, unlike the traditional distributed learning systems, the FL system does not have control over users' devices. For example, when a mobile phone is turned off or WiFi access is unavailable, the central server will lose connection to this device. When this happens during training, such a non-responding/inactive device, which is called a straggler, appears tremendously slower than the other devices. Unfortunately, since it has no control over the devices, the system can do nothing but waiting or ignoring the stragglers. Waiting for all the devices' response is obviously infeasible; it is thus impractical to require all the devices be active.

Third, the training data are non-iid[2], that is, a device's local data cannot be regarded as samples drawn from the overall distribution. The data available locally fail to represent the overall distribution. This does not only bring challenges to algorithm design but also make theoretical analysis much harder. While FedAvg actually works when the data are non-iid McMahan et al. (2017), FedAvg on non-iid data lacks theoretical guarantee even in convex optimization setting.

There have been much efforts developing convergence guarantees for FL algorithm based on the assumptions that (1) the data are iid and (2) all the devices are active. Khaled et al. (2019); Yu et al. (2019); Wang et al. (2019) made the latter assumption, while Zhou and Cong (2017); Stich (2018); Wang and Joshi (2018); Woodworth et al. (2018) made both assumptions. The two assumptions violates the second and third characters of FL. Previous algorithm Fedprox Sahu et al. (2018) doesn't require the two mentioned assumptions and incorporates FedAvg as a special case when the added proximal term vanishes. However, their theory fails to cover FedAvg.

**Notation.** Let $N$ be the total number of user devices and $K$ ($\leq N$) be the maximal number of devices that participate in every round's communication. Let $T$ be the total number of every device's SGDs, $E$ be the number of local iterations performed in a device between two communications, and thus $\frac{T}{E}$ is the number of communications.

**Contributions.** For strongly convex and smooth problems, we establish a convergence guarantee for FedAvg without making the two impractical assumptions: (1) the data are iid, and (2) all the devices are active. To the best of our knowledge, this work is the first to show the convergence rate of FedAvg without making the two assumptions.

We show in Theorem 1, 2, and 3 that FedAvg has $\mathcal{O}(\frac{1}{T})$ convergence rate. In particular, Theorem 3 shows that to attain a fixed precision $\epsilon$, the number of communications is

$$\frac{T}{E} = \mathcal{O}\left[\frac{1}{\epsilon}\left(\left(1 + \frac{1}{K}\right)EG^2 + \frac{\sum_{k=1}^{N} p_k^2 \sigma_k^2 + \Gamma + G^2}{E}\right)\right]. \tag{1}$$

Here, $G$, $\Gamma$, $p_k$, and $\sigma_k$ are problem-related constants defined in Section 3.1. The most interesting insight is that $E$ is a knob controlling the convergence rate: neither setting $E$ over-small ($E = 1$ makes FedAvg equivalent to SGD) nor setting $E$ over-large is good for the convergence.

This work also makes algorithmic contributions. We summarize the existing sampling[3] and averaging schemes for FedAvg (which do not have convergence bounds before this work) and propose a new scheme (see Table 1). We point out that a suitable sampling and averaging scheme is crucial for the convergence of FedAvg. To the best of our knowledge, we are the first to theoretically demonstrate

---

[1]In original paper (McMahan et al., 2017), $E$ epochs of SGD are performed in parallel. For theoretical analyses, we denote by $E$ the times of updates rather than epochs.

[2]Throughout this paper, "non-iid" means data are not identically distributed. More precisely, the data distributions in the $k$-th and $l$-th devices, denote $D_k$ and $D_l$, can be different.

[3]Throughout this paper, "sampling" refers to how the server chooses $K$ user devices and use their outputs for updating the model parameters. "Sampling" does not mean how a device randomly selects training samples.

Table 1: Sampling and averaging schemes for FedAvg. $\mathcal{S}_t \sim \mathcal{U}(N, K)$ means $\mathcal{S}_t$ is a size-$K$ subset uniformly sampled **without replacement** from $[N]$. $\mathcal{S}_t \sim \mathcal{W}(N, K, \mathbf{p})$ means $\mathcal{S}_t$ contains $K$ elements that are iid sampled **with replacement** from $[N]$ with probabilities $\{p_k\}$. In the latter scheme, $\mathcal{S}_t$ is not a set.

| Paper | Sampling | Averaging | Convergence rate |
|---|---|---|---|
| McMahan et al. (2017) | $\mathcal{S}_t \sim \mathcal{U}(N, K)$ | $\sum_{k \notin \mathcal{S}_t} p_k \mathbf{w}_t + \sum_{k \in \mathcal{S}_t} p_k \mathbf{w}_t^k$ | - |
| Sahu et al. (2018) | $\mathcal{S}_t \sim \mathcal{W}(N, K, \mathbf{p})$ | $\frac{1}{K} \sum_{k \in \mathcal{S}_t} \mathbf{w}_t^k$ | $\mathcal{O}(\frac{1}{T})$[5] |
| Ours | $\mathcal{S}_t \sim \mathcal{U}(N, K)$ | $\sum_{k \in \mathcal{S}_t} p_k \frac{N}{K} \mathbf{w}_t^k$ | $\mathcal{O}(\frac{1}{T})$[6] |

that FedAvg with certain schemes (see Table 1) can achieve $\mathcal{O}(\frac{1}{T})$ convergence rate in non-iid federated setting. We show that heterogeneity of training data and partial device participation slow down the convergence. We empirically verify our results through numerical experiments.

Our theoretical analysis requires the decay of learning rate (which is known to hinder the convergence rate.) Unfortunately, we show in Theorem 4 that the decay of learning rate is necessary for FedAvg with $E > 1$, even if full gradient descent is used.[4] If the learning rate is fixed to $\eta$ throughout, FedAvg would converge to a solution at least $\Omega(\eta(E-1))$ away from the optimal. To establish Theorem 4, we construct a specific $\ell_2$-norm regularized linear regression model which satisfies our strong convexity and smoothness assumptions.

**Paper organization.** In Section 2, we elaborate on FedAvg. In Section 3, we present our main convergence bounds for FedAvg. In Section 4, we construct a special example to show the necessity of learning rate decay. In Section 5, we discuss and compare with prior work. In Section 6, we conduct empirical study to verify our theories. All the proofs are left to the appendix.

## 2 FEDERATED AVERAGING (FEDAVG)

**Problem formulation.** In this work, we consider the following distributed optimization model:

$$\min_{\mathbf{w}} \left\{ F(\mathbf{w}) \triangleq \sum_{k=1}^{N} p_k F_k(\mathbf{w}) \right\}, \tag{2}$$

where $N$ is the number of devices, and $p_k$ is the weight of the $k$-th device such that $p_k \geq 0$ and $\sum_{k=1}^{N} p_k = 1$. Suppose the $k$-th device holds the $n_k$ training data: $x_{k,1}, x_{k,2}, \cdots, x_{k,n_k}$. The local objective $F_k(\cdot)$ is defined by

$$F_k(\mathbf{w}) \triangleq \frac{1}{n_k} \sum_{j=1}^{n_k} \ell(\mathbf{w}; x_{k,j}), \tag{3}$$

where $\ell(\cdot; \cdot)$ is a user-specified loss function.

**Algorithm description.** Here, we describe one around (say the $t$-th) of the *standard* FedAvg algorithm. First, the central server **broadcasts** the latest model, $\mathbf{w}_t$, to all the devices. Second, every device (say the $k$-th) lets $\mathbf{w}_t^k = \mathbf{w}_t$ and then performs $E$ ($\geq 1$) **local updates**:

$$\mathbf{w}_{t+i+1}^k \longleftarrow \mathbf{w}_{t+i}^k - \eta_{t+i} \nabla F_k(\mathbf{w}_{t+i}^k, \xi_{t+i}^k), i = 0, 1, \cdots, E - 1$$

where $\eta_{t+i}$ is the learning rate (a.k.a. step size) and $\xi_{t+i}^k$ is a sample uniformly chosen from the local data. Last, the server **aggregates** the local models, $\mathbf{w}_{t+E}^1, \cdots, \mathbf{w}_{t+E}^N$, to produce the new global model, $\mathbf{w}_{t+E}$. Because of the non-iid and partial device participation issues, the aggregation step can vary.

---

[4]It is well know that the full gradient descent (which is equivalent to FedAvg with $E = 1$ and full batch) do not require the decay of learning rate.

[5]The sampling scheme is proposed by Sahu et al. (2018) for FedAvg as a baseline, but this convergence rate is our contribution.

[6]The convergence relies on the assumption that data are balanced, i.e., $n_1 = n_2 = \cdots = n_N$. However, we can use a rescaling trick to get rid of this assumption. We will discuss this point later in Section 3.

**IID versus non-iid.** Suppose the data in the $k$-th device are i.i.d. sampled from the distribution $\mathcal{D}_k$. Then the overall distribution is a mixture of all local data distributions: $\mathcal{D} = \sum_{k=1}^{N} p_k \mathcal{D}_k$. The prior work Zhang et al. (2015a); Zhou and Cong (2017); Stich (2018); Wang and Joshi (2018); Woodworth et al. (2018) assumes the data are iid generated by or partitioned among the $N$ devices, that is, $\mathcal{D}_k = \mathcal{D}$ for all $k \in [N]$. However, real-world applications do not typically satisfy the iid assumption. One of our theoretical contributions is avoiding making the iid assumption.

**Full device participation.** The prior work Coppola (2015); Zhou and Cong (2017); Stich (2018); Yu et al. (2019); Wang and Joshi (2018); Wang et al. (2019) requires the full device participation in the aggregation step of `FedAvg`. In this case, the aggregation step performs

$$\mathbf{w}_{t+E} \;\longleftarrow\; \sum_{k=1}^{N} p_k \,\mathbf{w}_{t+E}^k.$$

Unfortunately, the full device participation requirement suffers from serious "straggler's effect" (which means everyone waits for the slowest) in real-world applications. For example, if there are thousands of users' devices in the FL system, there are always a small portion of devices offline. Full device participation means the central server must wait for these "stragglers", which is obviously unrealistic.

**Partial device participation.** This strategy is much more realistic because it does not require all the devices' output. We can set a threshold $K$ ($1 \le K < N$) and let the central server collect the outputs of the first $K$ responded devices. After collecting $K$ outputs, the server stops waiting for the rest; the $K+1$-th to $N$-th devices are regarded stragglers in this iteration. Let $\mathcal{S}_t$ ($|\mathcal{S}_t| = K$) be the set of the indices of the first $K$ responded devices in the $t$-th iteration. The aggregation step performs

$$\mathbf{w}_{t+E} \;\longleftarrow\; \frac{N}{K} \sum_{k \in \mathcal{S}_t} p_k \,\mathbf{w}_{t+E}^k.$$

It can be proved that $\frac{N}{K} \sum_{k \in \mathcal{S}_t} p_k$ equals one in expectation.

**Communication cost.** The `FedAvg` requires two rounds communications— one broadcast and one aggregation— per $E$ iterations. If $T$ iterations are performed totally, then the number of communications is $\lfloor \frac{2T}{E} \rfloor$. During the broadcast, the central server sends $\mathbf{w}_t$ to all the devices. During the aggregation, all or part of the $N$ devices sends its output, say $\mathbf{w}_{t+E}^k$, to the server.

## 3 CONVERGENCE ANALYSIS OF FEDAVG IN NON-IID SETTING

In this section, we show that `FedAvg` converges to the global optimum at a rate of $\mathcal{O}(1/T)$ for strongly convex and smooth functions and non-iid data. The main observation is that when the learning rate is sufficiently small, the effect of $E$ steps of local updates is similar to one step update with a larger learning rate. This coupled with appropriate sampling and averaging schemes would make each global update behave like an SGD update. Partial device participation ($K < N$) only makes the averaged sequence $\{\mathbf{w}_t\}$ have a larger variance, which, however, can be controlled by learning rates. These imply the convergence property of `FedAvg` should not differ too much from SGD. Next, we will first give the convergence result with full device participation (i.e., $K = N$) and then extend this result to partial device participation (i.e., $K < N$).

### 3.1 NOTATION AND ASSUMPTIONS

We make the following assumptions on the functions $F_1, \cdots, F_N$. Assumption 1 and 2 are standard; typical examples are the $\ell_2$-norm regularized linear regression, logistic regression, and softmax classifier.

**Assumption 1.** $F_1, \cdots, F_N$ are all $L$-smooth: for all $\mathbf{v}$ and $\mathbf{w}$, $F_k(\mathbf{v}) \le F_k(\mathbf{w}) + (\mathbf{v} - \mathbf{w})^T \nabla F_k(\mathbf{w}) + \frac{L}{2} \|\mathbf{v} - \mathbf{w}\|_2^2$.

**Assumption 2.** $F_1, \cdots, F_N$ are all $\mu$-strongly convex: for all $\mathbf{v}$ and $\mathbf{w}$, $F_k(\mathbf{v}) \ge F_k(\mathbf{w}) + (\mathbf{v} - \mathbf{w})^T \nabla F_k(\mathbf{w}) + \frac{\mu}{2} \|\mathbf{v} - \mathbf{w}\|_2^2$.

Assumptions 3 and 4 have been made by the works Zhang et al. (2013); Stich (2018); Stich et al. (2018); Yu et al. (2019).

**Assumption 3.** *Let $\xi_t^k$ be sampled from the $k$-th device's local data uniformly at random. The variance of stochastic gradients in each device is bounded: $\mathbb{E} \left\| \nabla F_k(\mathbf{w}_t^k, \xi_t^k) - \nabla F_k(\mathbf{w}_t^k) \right\|^2 \le \sigma_k^2$ for $k = 1, \cdots, N$.*

**Assumption 4.** *The expected squared norm of stochastic gradients is uniformly bounded, i.e., $\mathbb{E} \left\| \nabla F_k(\mathbf{w}_t^k, \xi_t^k) \right\|^2 \le G^2$ for all $k = 1, \cdots, N$ and $t = 0, \cdots, T - 1$.*

**Quantifying the degree of non-iid (heterogeneity).** Let $F^*$ and $F_k^*$ be the minimum values of $F$ and $F_k$, respectively. We use the term $\Gamma = F^* - \sum_{k=1}^N p_k F_k^*$ for quantifying the degree of non-iid. If the data are iid, then $\Gamma$ obviously goes to zero as the number of samples grows. If the data are non-iid, then $\Gamma$ is nonzero, and its magnitude reflects the heterogeneity of the data distribution.

## 3.2 CONVERGENCE RESULT: FULL DEVICE PARTICIPATION

Here we analyze the case that all the devices participate in the aggregation step; see Section 2 for the algorithm description. Let the FedAvg algorithm terminate after $T$ iterations and return $\mathbf{w}_T$ as the solution. We always require $T$ is evenly divisible by $E$ so that FedAvg can output $\mathbf{w}_T$ as expected.

**Theorem 1.** *Let Assumptions 1 to 4 hold and $L, \mu, \sigma_k, G$ be defined therein. Choose $\kappa = \frac{L}{\mu}$, $\gamma = \max\{8\kappa, E\}$ and the learning rate $\eta_t = \frac{2}{\mu(\gamma+t)}$. Then FedAvg with full device participation satisfies*

$$\mathbb{E}\left[F(\mathbf{w}_T)\right] - F^* \le \frac{2\kappa}{\gamma + T} \left( \frac{B}{\mu} + 2L\|\mathbf{w}_0 - \mathbf{w}^*\|^2 \right), \tag{4}$$

*where*

$$B = \sum_{k=1}^N p_k^2 \sigma_k^2 + 6L\Gamma + 8(E-1)^2 G^2. \tag{5}$$

## 3.3 CONVERGENCE RESULT: PARTIAL DEVICE PARTICIPATION

As discussed in Section 2, partial device participation has more practical interest than full device participation. Let the set $\mathcal{S}_t$ ($\subset [N]$) index the active devices in the $t$-th iteration. To establish the convergence bound, we need to make assumptions on $\mathcal{S}_t$.

Assumption 5 assumes the $K$ indices are selected from the distribution $p_k$ independently and with replacement. The aggregation step is simply averaging. This is first proposed in (Sahu et al., 2018), but they did not provide theoretical analysis.

**Assumption 5** (Scheme I). *Assume $\mathcal{S}_t$ contains a subset of $K$ indices randomly selected with replacement according to the sampling probabilities $p_1, \cdots, p_N$. The aggregation step of FedAvg performs $\mathbf{w}_t \longleftarrow \frac{1}{K} \sum_{k \in \mathcal{S}_t} \mathbf{w}_t^k$.*

**Theorem 2.** *Let Assumptions 1 to 4 hold and $L, \mu, \sigma_k, G$ be defined therein. Let $\kappa, \gamma, \eta_t$, and $B$ be defined in Theorem 1. Let Assumption 5 hold and define $C = \frac{4}{K}E^2G^2$. Then*

$$\mathbb{E}\left[F(\mathbf{w}_T)\right] - F^* \le \frac{2\kappa}{\gamma + T} \left( \frac{B + C}{\mu} + 2L\|\mathbf{w}_0 - \mathbf{w}^*\|^2 \right). \tag{6}$$

Alternatively, we can select $K$ indices from $[N]$ uniformly at random without replacement. As a consequence, we need a different aggregation strategy. Assumption 6 assumes the $K$ indices are selected uniformly without replacement and the aggregation step is the same as in Section 2. However, to guarantee convergence, we require an additional assumption of balanced data.

**Assumption 6** (Scheme II). *Assume $\mathcal{S}_t$ contains a subset of $K$ indices uniformly sampled from $[N]$ without replacement. Assume the data is balanced in the sense that $p_1 = \cdots = p_N = \frac{1}{N}$. The aggregation step of FedAvg performs $\mathbf{w}_t \longleftarrow \frac{N}{K} \sum_{k \in \mathcal{S}_t} p_k \mathbf{w}_t^k$.*

**Theorem 3.** *Replace Assumption 5 by Assumption 6 and $C$ by $C = \frac{N-K}{N-1} \frac{4}{K} E^2 G^2$. Then the same bound in Theorem 2 holds.*

Scheme II requires $p_1 = \cdots = p_N = \frac{1}{N}$ which obviously violates the unbalance nature of FL. Fortunately, this can be addressed by the following transformation. Let $\widetilde{F}_k(\mathbf{w}) = p_k N F_k(\mathbf{w})$ be a scaled local objective $F_k$. Then the global objective becomes a simple average of all scaled local objectives:

$$F(\mathbf{w}) = \sum_{k=1}^{N} p_k F_k(\mathbf{w}) = \frac{1}{N} \sum_{k=1}^{N} \widetilde{F}_k(\mathbf{w}).$$

Theorem 3 still holds if $L, \mu, \sigma_k, G$ are replaced by $\widetilde{L} \triangleq \nu L$, $\widetilde{\mu} \triangleq \varsigma \mu$, $\widetilde{\sigma}_k = \sqrt{\nu} \sigma$, and $\widetilde{G} = \sqrt{\nu} G$, respectively. Here, $\nu = N \cdot \max_k p_k$ and $\varsigma = N \cdot \min_k p_k$.

## 3.4 Discussions

**Choice of $E$.** Since $\|\mathbf{w}_0 - \mathbf{w}^*\|^2 \leq \frac{4}{\mu^2} G^2$ for $\mu$-strongly convex $F$, the dominating term in eqn. (6) is

$$\mathcal{O}\left( \frac{\sum_{k=1}^{N} p_k^2 \sigma_k^2 + L\Gamma + \left(1 + \frac{1}{K}\right) E^2 G^2 + \kappa G^2}{\mu T} \right). \tag{7}$$

Let $T_\epsilon$ denote the number of required steps for `FedAvg` to achieve an $\epsilon$ accuracy. It follows from eqn. (7) that the number of required communication rounds is roughly

$$\frac{T_\epsilon}{E} \propto \left(1 + \frac{1}{K}\right) E G^2 + \frac{\sum_{k=1}^{N} p_k^2 \sigma_k^2 + L\Gamma + \kappa G^2}{E}. \tag{8}$$

Thus, $\frac{T_\epsilon}{E}$ is a function of $E$ that first decreases and then increases, which implies that over-small or over-large $E$ may lead to high communication cost and that the optimal $E$ exists.

Stich (2018) showed that if the data are iid, then $E$ can be set to $\mathcal{O}(\sqrt{T})$. However, this setting does not work if the data are non-iid. Theorem 1 implies that $E$ must not exceed $\Omega(\sqrt{T})$; otherwise, convergence is not guaranteed. Here we give an intuitive explanation. If $E$ is set big, then $\mathbf{w}_t^k$ can converge to the minimizer of $F_k$, and thus `FedAvg` becomes the one-shot average Zhang et al. (2013) of the local solutions. If the data are non-iid, the one-shot averaging does not work because weighted average of the minimizers of $F_1, \cdots, F_N$ can be very different from the minimizer of $F$.

**Choice of $K$.** Stich (2018) showed that if the data are iid, the convergence rate improves substantially as $K$ increases. However, under the non-iid setting, the convergence rate has a weak dependence on $K$, as we show in Theorems 2 and 3. This implies `FedAvg` is unable to achieve linear speedup. We have empirically observed this phenomenon (see Section 6). Thus, in practice, the participation ratio $\frac{K}{N}$ can be set small to alleviate the straggler's effect without affecting the convergence rate.

**Choice of sampling schemes.** We considered two sampling and averaging schemes in Theorems 2 and 3. Scheme I selects $K$ devices according to the probabilities $p_1, \cdots, p_N$ with replacement. The non-uniform sampling results in faster convergence than uniform sampling, especially when $p_1, \cdots, p_N$ are highly non-uniform. If the system can choose to activate any of the $N$ devices at any time, then Scheme I should be used.

However, oftentimes the system has no control over the sampling; instead, the server simply uses the first $K$ returned results for the update. In this case, we can assume the $K$ devices are uniformly sampled from all the $N$ devices and use Theorem 3 to guarantee the convergence. If $p_1, \cdots, p_N$ are highly non-uniform, then $\nu = N \cdot \max_k p_k$ is big and $\varsigma = N \cdot \min_k p_k$ is small, which makes the convergence of `FedAvg` slow. This point of view is empirically verified in our experiments.

## 4 Necessity of Learning Rate Decay

In this section, we point out that diminishing learning rates are crucial for the convergence of `FedAvg` in the non-iid setting. Specifically, we establish the following theorem by constructing a ridge regression model (which is strongly convex and smooth).

**Theorem 4.** *We artificially construct a strongly convex and smooth distributed optimization problem. With full batch size, $E > 1$, and any fixed step size,* `FedAvg` *will converge to sub-optimal points. Specifically, let $\tilde{\mathbf{w}}^*$ be the solution produced by* `FedAvg` *with a small enough and constant $\eta$, and $\mathbf{w}^*$ the optimal solution. Then we have*

$$\|\tilde{\mathbf{w}}^* - \mathbf{w}^*\|_2 = \Omega((E-1)\eta) \cdot \|\mathbf{w}^*\|_2.$$

*where we hide some problem dependent constants.*

Theorem 4 and its proof provide several implications. First, the decay of learning rate is necessary of `FedAvg`. On the one hand, Theorem 1 shows with $E > 1$ and a decaying learning rate, `FedAvg` converges to the optimum. On the other hand, Theorem 4 shows that with $E > 1$ and any fixed learning rate, `FedAvg` does not converges to the optimum.

Second, `FedAvg` behaves very differently from gradient descent. Note that `FedAvg` with $E = 1$ and full batch size is exactly the `Full Gradient Descent`; with a proper and fixed learning rate, its global convergence to the optimum is guaranteed Nesterov (2013). However, Theorem 4 shows that `FedAvg` with $E > 1$ and full batch size cannot possibly converge to the optimum. This conclusion doesn't contradict with Theorem 1 in Khaled et al. (2019), which, when translated into our case, asserts that $\tilde{\mathbf{w}}^*$ will locate in the neighborhood of $\mathbf{w}^*$ with a constant learning rate.

Third, Theorem 4 shows the requirement of learning rate decay is not an artifact of our analysis; instead, it is inherently required by `FedAvg`. An explanation is that constant learning rates, combined with $E$ steps of possibly-biased local updates, form a sub-optimal update scheme, but a diminishing learning rate can gradually eliminate such bias.

The efficiency of `FedAvg` principally results from the fact that it performs several update steps on a local model before communicating with other workers, which saves communication. Diminishing step sizes often hinders fast convergence, which may counteract the benefit of performing multiple local updates. Theorem 4 motivates more efficient alternatives to `FedAvg`.

## 5   RELATED WORK

Federated learning (FL) was first proposed by McMahan et al. (2017) for collaboratively learning a model without collecting users' data. The research work on FL is focused on the communication-efficiency Konevcnỳ et al. (2016); McMahan et al. (2017); Sahu et al. (2018); Smith et al. (2017) and data privacy Bagdasaryan et al. (2018); Bonawitz et al. (2017); Geyer et al. (2017); Hitaj et al. (2017); Melis et al. (2019). This work is focused on the communication-efficiency issue.

`FedAvg`, a synchronous distributed optimization algorithm, was proposed by McMahan et al. (2017) as an effective heuristic. Sattler et al. (2019); Zhao et al. (2018) studied the non-iid setting, however, they do not have convergence rate. A contemporaneous and independent work Xie et al. (2019) analyzed asynchronous `FedAvg`; while they did not require iid data, their bound do not guarantee convergence to saddle point or local minimum. Sahu et al. (2018) proposed a federated optimization framework called `FedProx` to deal with statistical heterogeneity and provided the convergence guarantees in non-iid setting. `FedProx` adds a proximal term to each local objective. When these proximal terms vanish, `FedProx` is reduced to `FedAvg`. However, their convergence theory requires the proximal terms always exist and hence fails to cover `FedAvg`.

When data are iid distributed and all devices are active, `FedAvg` is referred to as `LocalSGD`. Due to the two assumptions, theoretical analysis of `LocalSGD` is easier than `FedAvg`. Stich (2018) demonstrated `LocalSGD` provably achieves the same linear speedup with strictly less communication for strongly-convex stochastic optimization. Coppola (2015); Zhou and Cong (2017); Wang and Joshi (2018) studied `LocalSGD` in the non-convex setting and established convergence results. Yu et al. (2019); Wang et al. (2019) recently analyzed `LocalSGD` for non-convex functions in heterogeneous settings. In particular, Yu et al. (2019) demonstrated `LocalSGD` also achieves $\mathcal{O}(1/\sqrt{NT})$ convergence (i.e., linear speedup) for non-convex optimization. Lin et al. (2018) empirically shows variants of `LocalSGD` increase training efficiency and improve the generalization performance of large batch sizes while reducing communication. For `LocalGD` on non-iid data (as opposed to `LocalSGD`), the best result is by the contemporaneous work (but slightly later than our first version) (Khaled et al., 2019). Khaled et al. (2019) used fixed learning rate $\eta$ and showed $\mathcal{O}(\frac{1}{T})$

convergence to a point $\mathcal{O}(\eta^2 E^2)$ away from the optimal. In fact, the suboptimality is due to their fixed learning rate. As we show in Theorem 4, using a fixed learning rate $\eta$ throughout, the solution by LocalGD is at least $\Omega((E-1)\eta)$ away from the optimal.

If the data are iid, distributed optimization can be efficiently solved by the second-order algorithms Mahajan et al. (2018); Reddi et al. (2016); Shamir et al. (2014); Shusen Wang et al. (2018); Zhang and Lin (2015) and the one-shot methods Lee et al. (2017); Lin et al. (2017); Wang (2019); Zhang et al. (2013; 2015b). The primal-dual algorithms Hong et al. (2018); Smith et al. (2016; 2017) are more generally applicable and more relevant to FL.

## 6   NUMERICAL EXPERIMENTS

**Models and datasets**   We examine our theoretical results on a logistic regression with weight decay $\lambda = 1e - 4$. This is a stochastic convex optimization problem. We distribute MNIST dataset (LeCun et al., 1998) among $N = 100$ workers in a non-iid fashion such that each device contains samples of only two digits. We further obtain two datasets: mnist balanced and mnist unbalanced. The former is balanced such that the number of samples in each device is the same, while the latter is highly unbalanced with the number of samples among devices following a power law. To manipulate heterogeneity more precisely, we synthesize unbalanced datasets following the setup in Sahu et al. (2018) and denote it as synthetic($\alpha$, $\beta$) where $\alpha$ controls how much local models differ from each other and $\beta$ controls how much the local data at each device differs from that of other devices. We obtain two datasets: synthetic(0,0) and synthetic(1,1). Details can be found in Appendix D.

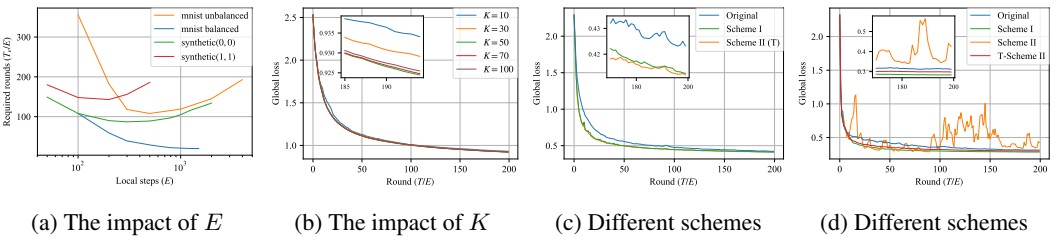

(a) The impact of $E$     (b) The impact of $K$     (c) Different schemes     (d) Different schemes

Figure 1: (a) To obtain an $\epsilon$ accuracy, the required rounds first decrease and then increase when we increase the local steps $E$. (b) In Synthetic(0,0) dataset, decreasing the numbers of active devices each round has little effect on the convergence process. (c) In mnist balanced dataset, Scheme I slightly outperforms Scheme II. They both performs better than the original scheme. Here transformed Scheme II coincides with Scheme II due to the balanced data. (d) In mnist unbalanced dataset, Scheme I performs better than Scheme II and the original scheme. Scheme II suffers from instability while transformed Scheme II has a lower convergence rate.

**Experiment settings**   For all experiments, we initialize all runnings with $\mathbf{w}_0 = 0$. In each round, all selected devices run $E$ steps of SGD in parallel. We decay the learning rate at the end of each round by the following scheme $\eta_t = \frac{\eta_0}{1+t}$, where $\eta_0$ is chosen from the set $\{1, 0.1, 0.01\}$. We evaluate the averaged model after each global synchronization on the corresponding global objective. For fair comparison, we control all randomness in experiments so that the set of activated devices is the same across all different algorithms on one configuration.

**Impact of $E$**   We expect that $T_\epsilon / E$, the required communication round to achieve curtain accuracy, is a hyperbolic finction of $E$ as equ (8) indicates. Intuitively, a small $E$ means a heavy communication burden, while a large $E$ means a low convergence rate. One needs to trade off between communication efficiency and fast convergence. We empirically observe this phenomenon on unbalanced datasets in Figure 1a. The reason why the phenomenon does not appear in mnist balanced dataset requires future investigations.

**Impact of** $K$   Our theory suggests that a larger $K$ may slightly accelerate convergence since $T_\epsilon/E$ contains a term $\mathcal{O}\left(\frac{EG^2}{K}\right)$. Figure 1b shows that $K$ has limited influence on the convergence of FedAvg in synthetic(0,0) dataset. It reveals that the curve of a large enough $K$ is slightly better. We observe similar phenomenon among the other three datasets and attach additional results in Appendix D. This justifies that when the variance resulting sampling is not too large (i.e., $B \gg C$), one can use a small number of devices without severely harming the training process, which also removes the need to sample as many devices as possible in convex federated optimization.

**Effect of sampling and averaging schemes.**   We compare four schemes among four federated datasets. Since the original scheme involves a history term and may be conservative, we carefully set the initial learning rate for it. Figure 1c indicates that when data are balanced, Schemes I and II achieve nearly the same performance, both better than the original scheme. Figure 1d shows that when the data are unbalanced, i.e., $p_k$'s are uneven, Scheme I performs the best. Scheme II suffers from some instability in this case. This is not contradictory with our theory since we don't guarantee the convergence of Scheme II when data is unbalanced. As expected, transformed Scheme II performs stably at the price of a lower convergence rate. Compared to Scheme I, the original scheme converges at a slower speed even if its learning rate is fine tuned. All the results show the crucial position of appropriate sampling and averaging schemes for FedAvg.

## 7 CONCLUSION

Federated learning becomes increasingly popular in machine learning and optimization communities. In this paper we have studied the convergence of FedAvg, a heuristic algorithm suitable for federated setting. We have investigated the influence of sampling and averaging schemes. We have provided theoretical guarantees for two schemes and empirically demonstrated their performances. Our work sheds light on theoretical understanding of FedAvg and provides insights for algorithm design in realistic applications. Though our analyses are constrained in convex problems, we hope our insights and proof techniques can inspire future work.

## ACKNOWLEDGEMENTS

Li, Yang and Zhang have been supported by the National Natural Science Foundation of China (No. 11771002 and 61572017), Beijing Natural Science Foundation (Z190001), the Key Project of MOST of China (No. 2018AAA0101000), and Beijing Academy of Artificial Intelligence (BAAI).

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

# A   PROOF OF THEOREM 1

We analyze `FedAvg` in the setting of full device participation in this section.

## A.1   ADDITIONAL NOTATION

Let $\mathbf{w}_t^k$ be the model parameter maintained in the $k$-th device at the $t$-th step. Let $\mathcal{I}_E$ be the set of global synchronization steps, i.e., $\mathcal{I}_E = \{nE \mid n = 1, 2, \cdots\}$. If $t + 1 \in \mathcal{I}_E$, i.e., the time step to communication, `FedAvg` activates all devices. Then the update of `FedAvg` with partial devices active can be described as

$$\mathbf{v}_{t+1}^k = \mathbf{w}_t^k - \eta_t \nabla F_k(\mathbf{w}_t^k, \xi_t^k), \tag{9}$$

$$\mathbf{w}_{t+1}^k = \begin{cases} \mathbf{v}_{t+1}^k & \text{if } t + 1 \notin \mathcal{I}_E, \\ \sum_{k=1}^N p_k \mathbf{v}_{t+1}^k & \text{if } t + 1 \in \mathcal{I}_E. \end{cases} \tag{10}$$

Here, an additional variable $\mathbf{v}_{t+1}^k$ is introduced to represent the immediate result of one step SGD update from $\mathbf{w}_t^k$. We interpret $\mathbf{w}_{t+1}^k$ as the parameter obtained after communication steps (if possible).

In our analysis, we define two virtual sequences $\overline{\mathbf{v}}_t = \sum_{k=1}^N p_k \mathbf{v}_t^k$ and $\overline{\mathbf{w}}_t = \sum_{k=1}^N p_k \mathbf{w}_t^k$. This is motivated by (Stich, 2018). $\overline{\mathbf{v}}_{t+1}$ results from an single step of SGD from $\overline{\mathbf{w}}_t$. When $t + 1 \notin \mathcal{I}_E$, both are inaccessible. When $t + 1 \in \mathcal{I}_E$, we can only fetch $\overline{\mathbf{w}}_{t+1}$. For convenience, we define $\overline{\mathbf{g}}_t = \sum_{k=1}^N p_k \nabla F_k(\mathbf{w}_t^k)$ and $\mathbf{g}_t = \sum_{k=1}^N p_k \nabla F_k(\mathbf{w}_t^k, \xi_t^k)$. Therefore, $\overline{\mathbf{v}}_{t+1} = \overline{\mathbf{w}}_t - \eta_t \mathbf{g}_t$ and $\mathbb{E}\mathbf{g}_t = \overline{\mathbf{g}}_t$.

## A.2   KEY LEMMAS

To convey our proof clearly, it would be necessary to prove certain useful lemmas. We defer the proof of these lemmas to latter section and focus on proving the main theorem.

**Lemma 1** (Results of one step SGD). *Assume Assumption 1 and 2. If $\eta_t \leq \frac{1}{4L}$, we have*

$$\mathbb{E}\left\|\overline{\mathbf{v}}_{t+1} - \mathbf{w}^\star\right\|^2 \leq (1 - \eta_t\mu)\mathbb{E}\left\|\overline{\mathbf{w}}_t - \mathbf{w}^\star\right\|^2 + \eta_t^2\mathbb{E}\left\|\mathbf{g}_t - \overline{\mathbf{g}}_t\right\|^2 + 6L\eta_t^2\Gamma + 2\mathbb{E}\sum_{k=1}^N p_k\left\|\overline{\mathbf{w}}_t - \mathbf{w}_k^t\right\|^2$$

*where $\Gamma = F^* - \sum_{k=1}^N p_k F_k^\star \geq 0$.*

**Lemma 2** (Bounding the variance). *Assume Assumption 3 holds. It follows that*

$$\mathbb{E}\left\|\mathbf{g}_t - \overline{\mathbf{g}}_t\right\|^2 \leq \sum_{k=1}^N p_k^2\sigma_k^2.$$

**Lemma 3** (Bounding the divergence of $\{\mathbf{w}_t^k\}$). *Assume Assumption 4, that $\eta_t$ is non-increasing and $\eta_t \leq 2\eta_{t+E}$ for all $t \geq 0$. It follows that*

$$\mathbb{E}\left[\sum_{k=1}^N p_k\left\|\overline{\mathbf{w}}_t - \mathbf{w}_k^t\right\|^2\right] \leq 4\eta_t^2(E-1)^2G^2.$$

### A.3 Completing the Proof of Theorem 1

*Proof.* It is clear that no matter whether $t + 1 \in \mathcal{I}_E$ or $t + 1 \notin \mathcal{I}_E$, we always have $\overline{\mathbf{w}}_{t+1} = \overline{\mathbf{v}}_{t+1}$. Let $\Delta_t = \mathbb{E} \left\| \overline{\mathbf{w}}_{t+1} - \mathbf{w}^\star \right\|^2$. From Lemma 1, Lemma 2 and Lemma 3, it follows that

$$\Delta_{t+1} \le (1 - \eta_t \mu) \Delta_t + \eta_t^2 B \tag{11}$$

where

$$B = \sum_{k=1}^{N} p_k^2 \sigma_k^2 + 6L\Gamma + 8(E-1)^2 G^2.$$

For a diminishing stepsize, $\eta_t = \frac{\beta}{t+\gamma}$ for some $\beta > \frac{1}{\mu}$ and $\gamma > 0$ such that $\eta_1 \le \min\{\frac{1}{\mu}, \frac{1}{4L}\} = \frac{1}{4L}$ and $\eta_t \le 2\eta_{t+E}$. We will prove $\Delta_t \le \frac{v}{\gamma+t}$ where $v = \max\left\{ \frac{\beta^2 B}{\beta\mu - 1}, (\gamma + 1)\Delta_1 \right\}$.

We prove it by induction. Firstly, the definition of $v$ ensures that it holds for $t = 1$. Assume the conclusion holds for some $t$, it follows that

$$\begin{aligned}
\Delta_{t+1} &\le (1 - \eta_t \mu)\Delta_t + \eta_t^2 B \\
&= \left(1 - \frac{\beta\mu}{t+\gamma}\right) \frac{v}{t+\gamma} + \frac{\beta^2 B}{(t+\gamma)^2} \\
&= \frac{t+\gamma-1}{(t+\gamma)^2} v + \left[ \frac{\beta^2 B}{(t+\gamma)^2} - \frac{\beta\mu - 1}{(t+\gamma)^2} v \right] \\
&\le \frac{v}{t+\gamma+1}.
\end{aligned}$$

Then by the strong convexity of $F(\cdot)$,

$$\mathbb{E}[F(\overline{\mathbf{w}}_t)] - F^* \le \frac{L}{2}\Delta_t \le \frac{L}{2} \frac{v}{\gamma + t}.$$

Specifically, if we choose $\beta = \frac{2}{\mu}, \gamma = \max\{8\frac{L}{\mu} - 1, E\}$ and denote $\kappa = \frac{L}{\mu}$, then $\eta_t = \frac{2}{\mu} \frac{1}{\gamma+t}$ and

$$\mathbb{E}[F(\overline{\mathbf{w}}_t)] - F^* \le \frac{2\kappa}{\gamma + t} \left( \frac{B}{\mu} + 2L\Delta_1 \right).$$

$\square$

### A.4 Deferred Proofs of Key Lemmas

*Proof of Lemma 1.* Notice that $\overline{\mathbf{v}}_{t+1} = \overline{\mathbf{w}}_t - \eta_t \mathbf{g}_t$, then

$$\begin{aligned}
\left\| \overline{\mathbf{v}}_{t+1} - \mathbf{w}^\star \right\|^2 &= \left\| \overline{\mathbf{w}}_t - \eta_t \mathbf{g}_t - \mathbf{w}^\star - \eta_t \overline{\mathbf{g}}_t + \eta_t \overline{\mathbf{g}}_t \right\|^2 \\
&= \underbrace{\left\| \overline{\mathbf{w}}_t - \mathbf{w}^\star - \eta_t \overline{\mathbf{g}}_t \right\|^2}_{A_1} + \underbrace{2\eta_t \left\langle \overline{\mathbf{w}}_t - \mathbf{w}^\star - \eta_t \overline{\mathbf{g}}_t, \overline{\mathbf{g}}_t - \mathbf{g}_t \right\rangle}_{A_2} + \eta_t^2 \left\| \mathbf{g}_t - \overline{\mathbf{g}}_t \right\|^2 \quad (12)
\end{aligned}$$

Note that $\mathbb{E}A_2 = 0$. We next focus on bounding $A_1$. Again we split $A_1$ into three terms:

$$\left\| \overline{\mathbf{w}}_t - \mathbf{w}^\star - \eta_t \overline{\mathbf{g}}_t \right\|^2 = \left\| \overline{\mathbf{w}}_t - \mathbf{w}^\star \right\|^2 \underbrace{-2\eta_t \left\langle \overline{\mathbf{w}}_t - \mathbf{w}^\star, \overline{\mathbf{g}}_t \right\rangle}_{B_1} + \underbrace{\eta_t^2 \left\| \overline{\mathbf{g}}_t \right\|^2}_{B_2} \tag{13}$$

From the the $L$-smoothness of $F_k(\cdot)$, it follows that

$$\left\| \nabla F_k \left( \mathbf{w}_t^k \right) \right\|^2 \le 2L \left( F_k \left( \mathbf{w}_t^k \right) - F_k^\star \right). \tag{14}$$

By the convexity of $\|\cdot\|^2$ and eqn. (14), we have

$$B_2 = \eta_t^2 \left\| \overline{\mathbf{g}}_t \right\|^2 \le \eta_t^2 \sum_{k=1}^{N} p_k \left\| \nabla F_k \left( \mathbf{w}_t^k \right) \right\|^2 \le 2L\eta_t^2 \sum_{k=1}^{N} p_k \left( F_k(\mathbf{w}_t^k) - F_k^* \right).$$

Note that

$$B_1 = -2\eta_t \langle \overline{\mathbf{w}}_t - \mathbf{w}^\star, \overline{\mathbf{g}}_t \rangle = -2\eta_t \sum_{k=1}^{N} p_k \langle \overline{\mathbf{w}}_t - \mathbf{w}^\star, \nabla F_k(\mathbf{w}_t^k) \rangle$$

$$= -2\eta_t \sum_{k=1}^{N} p_k \langle \overline{\mathbf{w}}_t - \mathbf{w}_t^k, \nabla F_k(\mathbf{w}_t^k) \rangle - 2\eta_t \sum_{k=1}^{N} p_k \langle \mathbf{w}_t^k - \mathbf{w}^\star, \nabla F_k(\mathbf{w}_t^k) \rangle. \qquad (15)$$

By Cauchy-Schwarz inequality and AM-GM inequality, we have

$$-2 \langle \overline{\mathbf{w}}_t - \mathbf{w}_t^k, \nabla F_k\left(\mathbf{w}_t^k\right) \rangle \le \frac{1}{\eta_t} \left\| \overline{\mathbf{w}}_t - \mathbf{w}_t^k \right\|^2 + \eta_t \left\| \nabla F_k\left(\mathbf{w}_t^k\right) \right\|^2. \qquad (16)$$

By the $\mu$-strong convexity of $F_k(\cdot)$, we have

$$-\langle \mathbf{w}_t^k - \mathbf{w}^\star, \nabla F_k\left(\mathbf{w}_t^k\right) \rangle \le -\left(F_k\left(\mathbf{w}_t^k\right) - F_k(\mathbf{w}^*)\right) - \frac{\mu}{2} \left\| \mathbf{w}_t^k - \mathbf{w}^\star \right\|^2. \qquad (17)$$

By combining eqn. (13), eqn. (15), eqn. (16) and eqn. (17), it follows that

$$A_1 = \left\| \overline{\mathbf{w}}_t - \mathbf{w}^\star - \eta_t \overline{\mathbf{g}}_t \right\|^2 \le \left\| \overline{\mathbf{w}}_t - \mathbf{w}^\star \right\|^2 + 2L\eta_t^2 \sum_{k=1}^{N} p_k \left(F_k(\mathbf{w}_t^k) - F_k^*\right)$$

$$+ \eta_t \sum_{k=1}^{N} p_k \left( \frac{1}{\eta_t} \left\| \overline{\mathbf{w}}_t - \mathbf{w}_k^t \right\|^2 + \eta_t \left\| \nabla F_k\left(\mathbf{w}_t^k\right) \right\|^2 \right)$$

$$- 2\eta_t \sum_{k=1}^{N} p_k \left( F_k\left(\mathbf{w}_t^k\right) - F_k(\mathbf{w}^*) + \frac{\mu}{2} \left\| \mathbf{w}_t^k - \mathbf{w}^\star \right\|^2 \right)$$

$$= (1 - \mu\eta_t) \left\| \overline{\mathbf{w}}_t - \mathbf{w}^\star \right\|^2 + \sum_{k=1}^{N} p_k \left\| \overline{\mathbf{w}}_t - \mathbf{w}_k^t \right\|^2$$

$$+ \underbrace{4L\eta_t^2 \sum_{k=1}^{N} p_k \left(F_k(\mathbf{w}_t^k) - F_k^*\right) - 2\eta_t \sum_{k=1}^{N} p_k \left(F_k\left(\mathbf{w}_t^k\right) - F_k(\mathbf{w}^*)\right)}_{C}$$

where we use eqn. (14) again.

We next aim to bound $C$. We define $\gamma_t = 2\eta_t(1 - 2L\eta_t)$. Since $\eta_t \le \frac{1}{4L}$, $\eta_t \le \gamma_t \le 2\eta_t$. Then we split $C$ into two terms:

$$C = -2\eta_t(1 - 2L\eta_t) \sum_{k=1}^{N} p_k \left(F_k(\mathbf{w}_t^k) - F_k^*\right) + 2\eta_t \sum_{k=1}^{N} p_k \left(F_k(\mathbf{w}^*) - F_k^*\right)$$

$$= -\gamma_t \sum_{k=1}^{N} p_k \left(F_k(\mathbf{w}_t^k) - F^*\right) + (2\eta_t - \gamma_t) \sum_{k=1}^{N} p_k \left(F^* - F_k^*\right)$$

$$= \underbrace{-\gamma_t \sum_{k=1}^{N} p_k \left(F_k(\mathbf{w}_t^k) - F^*\right)}_{D} + 4L\eta_t^2 \Gamma$$

where in the last equation, we use the notation $\Gamma = \sum_{k=1}^{N} p_k \left(F^* - F_k^*\right) = F^* - \sum_{k=1}^{N} p_k F_k^*$.

To bound $D$, we have

$$\sum_{k=1}^{N} p_k \left(F_k(\mathbf{w}_t^k) - F^*\right) = \sum_{k=1}^{N} p_k \left(F_k(\mathbf{w}_t^k) - F_k(\overline{\mathbf{w}}_t)\right) + \sum_{k=1}^{N} p_k \left(F_k(\overline{\mathbf{w}}_t) - F^*\right)$$

$$\geq \sum_{k=1}^{N} p_k \left\langle \nabla F_k(\overline{\mathbf{w}}_t), \overline{\mathbf{w}}_t^k - \overline{\mathbf{w}}_t \right\rangle + (F(\overline{\mathbf{w}}_t) - F^*)$$

$$\geq -\frac{1}{2} \sum_{k=1}^{N} p_k \left[ \eta_t \left\| \nabla F_k(\overline{\mathbf{w}}_t) \right\|^2 + \frac{1}{\eta_t} \left\| \mathbf{w}_t^k - \overline{\mathbf{w}}_t \right\|^2 \right] + (F(\overline{\mathbf{w}}_t) - F^*)$$

$$\geq -\sum_{k=1}^{N} p_k \left[ \eta_t L \left(F_k(\overline{\mathbf{w}}_t) - F_k^*\right) + \frac{1}{2\eta_t} \left\| \mathbf{w}_t^k - \overline{\mathbf{w}}_t \right\|^2 \right] + (F(\overline{\mathbf{w}}_t) - F^*)$$

where the first inequality results from the convexity of $F_k(\cdot)$, the second inequality from AM-GM inequality and the third inequality from eqn. (14).

Therefore

$$C = \gamma_t \sum_{k=1}^{N} p_k \left[ \eta_t L \left(F_k(\overline{\mathbf{w}}_t) - F_k^*\right) + \frac{1}{2\eta_t} \left\| \mathbf{w}_t^k - \overline{\mathbf{w}}_t \right\|^2 \right] - \gamma_t \left(F(\overline{\mathbf{w}}_t) - F^*\right) + 4L\eta_t^2 \Gamma$$

$$= \gamma_t(\eta_t L - 1) \sum_{k=1}^{N} p_k \left(F_k(\overline{\mathbf{w}}_t) - F^*\right) + \left(4L\eta_t^2 + \gamma_t \eta_t L\right) \Gamma + \frac{\gamma_t}{2\eta_t} \sum_{k=1}^{N} p_k \left\| \mathbf{w}_t^k - \overline{\mathbf{w}}_t \right\|^2$$

$$\leq 6L\eta_t^2 \Gamma + \sum_{k=1}^{N} p_k \left\| \mathbf{w}_t^k - \overline{\mathbf{w}}_t \right\|^2$$

where in the last inequality, we use the following facts: (1) $\eta_t L - 1 \leq -\frac{3}{4} \leq 0$ and $\sum_{k=1}^{N} p_k \left(F_k(\overline{\mathbf{w}}_t) - F^*\right) = F(\overline{\mathbf{w}}_t) - F^* \geq 0$ (2) $\Gamma \geq 0$ and $4L\eta_t^2 + \gamma_t \eta_t L \leq 6\eta_t^2 L$ and (3) $\frac{\gamma_t}{2\eta_t} \leq 1$.

Recalling the expression of $A_1$ and plugging $C$ into it, we have

$$A_1 = \left\| \overline{\mathbf{w}}_t - \mathbf{w}^\star - \eta_t \overline{\mathbf{g}}_t \right\|^2$$

$$\leq (1 - \mu\eta_t) \left\| \overline{\mathbf{w}}_t - \mathbf{w}^\star \right\|^2 + 2 \sum_{k=1}^{N} p_k \left\| \overline{\mathbf{w}}_t - \mathbf{w}_k^t \right\|^2 + 6\eta_t^2 L\Gamma \tag{18}$$

Using eqn. (18) and taking expectation on both sides of eqn. (12), we erase the randomness from stochastic gradients, we complete the proof. □

*Proof of Lemma 2.* From Assumption 3, the variance of the stochastic gradients in device $k$ is bounded by $\sigma_k^2$, then

$$\mathbb{E} \left\| \mathbf{g}_t - \overline{\mathbf{g}}_t \right\|^2 = \mathbb{E} \left\| \sum_{k=1}^{N} p_k (\nabla F_k(\mathbf{w}_t^k, \xi_t^k) - \nabla F_k(\mathbf{w}_t^k)) \right\|^2,$$

$$= \sum_{k=1}^{N} p_k^2 \mathbb{E} \left\| \nabla F_k(\mathbf{w}_t^k, \xi_t^k) - \nabla F_k(\mathbf{w}_t^k) \right\|^2,$$

$$\leq \sum_{k=1}^{N} p_k^2 \sigma_k^2.$$

□

*Proof of Lemma 3.* Since `FedAvg` requires a communication each $E$ steps. Therefore, for any $t \geq 0$, there exists a $t_0 \leq t$, such that $t - t_0 \leq E - 1$ and $\mathbf{w}_{t_0}^k = \overline{\mathbf{w}}_{t_0}$ for all $k = 1, 2, \cdots, N$. Also, we use

the fact that $\eta_t$ is non-increasing and $\eta_{t_0} \leq 2\eta_t$ for all $t - t_0 \leq E - 1$, then

$$\mathbb{E} \sum_{k=1}^N p_k \left\| \overline{\mathbf{w}}_t - \mathbf{w}_t^k \right\|^2 = \mathbb{E} \sum_{k=1}^N p_k \left\| (\mathbf{w}_t^k - \overline{\mathbf{w}}_{t_0}) - (\overline{\mathbf{w}}_t - \overline{\mathbf{w}}_{t_0}) \right\|^2$$

$$\leq \mathbb{E} \sum_{k=1}^N p_k \left\| \mathbf{w}_t^k - \overline{\mathbf{w}}_{t_0} \right\|^2$$

$$\leq \sum_{k=1}^N p_k \mathbb{E} \sum_{t=t_0}^{t-1} (E-1)\eta_t^2 \left\| \nabla F_k(\mathbf{w}_t^k, \xi_t^k) \right\|^2$$

$$\leq \sum_{k=1}^N p_k \sum_{t=t_0}^{t-1} (E-1)\eta_{t_0}^2 G^2$$

$$\leq \sum_{k=1}^N p_k \eta_{t_0}^2 (E-1)^2 G^2$$

$$\leq 4\eta_t^2 (E-1)^2 G^2.$$

$\square$

# B   PROOFS OF THEOREMS 2 AND 3

We analyze `FedAvg` in the setting of partial device participation in this section.

## B.1   ADDITIONAL NOTATION

Recall that $\mathbf{w}_t^k$ is the model parameter maintained in the $k$-th device at the $t$-th step. $\mathcal{I}_E = \{nE \mid n = 1, 2, \cdots\}$ is the set of global synchronization steps. Unlike the setting in Appendix A, when it is the time to communicate, i.e., $t + 1 \in \mathcal{I}_E$, the scenario considered here is that `FedAvg` randomly activates a subset of devices according to some sampling schemes. Again, $\overline{\mathbf{g}}_t = \sum_{k=1}^N p_k \nabla F_k(\mathbf{w}_t^k)$ and $\mathbf{g}_t = \sum_{k=1}^N p_k F_k(\mathbf{w}_t^k, \xi_t^k)$. Therefore, $\overline{\mathbf{v}}_{t+1} = \overline{\mathbf{w}}_t - \eta_t \mathbf{g}_t$ and $\mathbb{E}\mathbf{g}_t = \overline{\mathbf{g}}_t$.

**Multiset selected.**   All sampling schemes can be divided into two groups, one with replacement and the other without replacement. For those with replacement, it is possible for a device to be activated several times in a round of communication, even though each activation is independent with the rest. We denote by $\mathcal{H}_t$ the multiset selected which allows any element to appear more than once. Note that $\mathcal{H}_t$ is only well defined for $t \in \mathcal{I}_E$. For convenience, we denote by $\mathcal{S}_t = \mathcal{H}_{N(t,E)}$ the most recent set of chosen devices where $N(t, E) = \max\{n \mid n \leq t, n \in \mathcal{I}_E\}$.

**Updating scheme.**   Limited to realistic scenarios (for communication efficiency and low straggler effect), FedAvg first samples a random multiset $\mathcal{S}_t$ of devices and then only perform updates on them. This make the analysis a little bit intricate, since $\mathcal{S}_t$ varies each $E$ steps. However, we can use a thought trick to circumvent this difficulty. We assume that `FedAvg` always activates **all devices** at the beginning of each round and then uses the parameters maintained in only a few sampled devices to produce the next-round parameter. It is clear that this updating scheme is equivalent to the original. Then the update of `FedAvg` with partial devices active can be described as: for all $k \in [N]$,

$$\mathbf{v}_{t+1}^k = \mathbf{w}_t^k - \eta_t \nabla F_k(\mathbf{w}_t^k, \xi_t^k), \tag{19}$$

$$\mathbf{w}_{t+1}^k = \begin{cases} \mathbf{v}_{t+1}^k & \text{if } t + 1 \notin \mathcal{I}_E, \\ \text{samples } \mathcal{S}_{t+1} \text{ and average } \{\mathbf{v}_{t+1}^k\}_{k \in \mathcal{S}_{t+1}} & \text{if } t + 1 \in \mathcal{I}_E. \end{cases} \tag{20}$$

**Sources of randomness.**   In our analysis, there are two sources of randomness. One results from the stochastic gradients and the other is from the random sampling of devices. All the analysis in Appendix A only involve the former. To distinguish them, we use the notation $\mathbb{E}_{\mathcal{S}_t}(\cdot)$, when we take expectation to erase the latter type of randomness.

### B.2 Key Lemmas

**Two schemes.** For full device participation, we always have $\overline{\mathbf{w}}_{t+1} = \overline{\mathbf{v}}_{t+1}$. This is true when $t + 1 \notin \mathcal{I}_E$ for partial device participation. When $t+1 \in \mathcal{I}_E$, we hope this relation establish in the sense of expectation. To that end, we require the sampling and averaging scheme to be unbiased in the sense that

$$\mathbb{E}_{\mathcal{S}_{t+1}}\overline{\mathbf{w}}_{t+1} = \overline{\mathbf{v}}_{t+1}.$$

We find two sampling and averaging schemes satisfying the requirement and provide convergence guarantees.

(I) The server establishes $\mathcal{S}_{t+1}$ by i.i.d. **with replacement** sampling an index $k \in \{1, \cdots, N\}$ with probabilities $p_1, \cdots, p_N$ for $K$ times. Hence $\mathcal{S}_{t+1}$ is a multiset which allows a element to occur more than once. Then the server averages the parameters by $\mathbf{w}_{t+1}^k = \frac{1}{K}\sum_{k \in \mathcal{S}_{t+1}} \mathbf{v}_{t+1}^k$. This is first proposed in (Sahu et al., 2018) but lacks theoretical analysis.

(II) The server samples $\mathcal{S}_{t+1}$ uniformly in a **without replacement** fashion. Hence each element in $\mathcal{S}_{t+1}$ only occurs once. Then server averages the parameters by $\mathbf{w}_{t+1}^k = \sum_{k \in \mathcal{S}_{t+1}} p_k \frac{N}{K}\mathbf{v}_{t+1}^k$. Note that when the $p_k$'s are not all the same, one cannot ensure $\sum_{k \in \mathcal{S}_{t+1}} p_k \frac{N}{K} = 1$.

**Unbiasedness and bounded variance.** Lemma 4 shows the mentioned two sampling and averaging schemes are unbiased. In expectation, the next-round parameter (i.e., $\overline{\mathbf{w}}_{t+1}$) is equal to the weighted average of parameters in **all devices** after SGD updates (i.e., $\overline{\mathbf{v}}_{t+1}$). However, the original scheme in (McMahan et al., 2017) (see Table 1) does not enjoy this property. But it is very similar to Scheme II except the averaging scheme. Hence our analysis cannot cover the original scheme.

Lemma 5 shows the expected difference between $\overline{\mathbf{v}}_{t+1}$ and $\overline{\mathbf{w}}_{t+1}$ is bounded. $\mathbb{E}_{\mathcal{S}_t}\|\overline{\mathbf{v}}_{t+1} - \overline{\mathbf{w}}_{t+1}\|^2$ is actually the variance of $\overline{\mathbf{w}}_{t+1}$.

**Lemma 4** (Unbiased sampling scheme). *If $t + 1 \in \mathcal{I}_E$, for Scheme I and Scheme II, we have*

$$\mathbb{E}_{\mathcal{S}_t}(\overline{\mathbf{w}}_{t+1}) = \overline{\mathbf{v}}_{t+1}.$$

**Lemma 5** (Bounding the variance of $\overline{\mathbf{w}}_t$). *For $t + 1 \in \mathcal{I}$, assume that $\eta_t$ is non-increasing and $\eta_t \leq 2\eta_{t+E}$ for all $t \geq 0$. We have the following results.*

*(1) For Scheme I, the expected difference between $\overline{\mathbf{v}}_{t+1}$ and $\overline{\mathbf{w}}_{t+1}$ is bounded by*

$$\mathbb{E}_{\mathcal{S}_t}\|\overline{\mathbf{v}}_{t+1} - \overline{\mathbf{w}}_{t+1}\|^2 \leq \frac{4}{K}\eta_t^2 E^2 G^2.$$

*(2) For Scheme II, assuming $p_1 = p_2 = \cdots = p_N = \frac{1}{N}$, the expected difference between $\overline{\mathbf{v}}_{t+1}$ and $\overline{\mathbf{w}}_{t+1}$ is bounded by*

$$\mathbb{E}_{\mathcal{S}_t}\|\overline{\mathbf{v}}_{t+1} - \overline{\mathbf{w}}_{t+1}\|^2 \leq \frac{N-K}{N-1}\frac{4}{K}\eta_t^2 E^2 G^2.$$

### B.3 Completing the Proof of Theorem 2 and 3

*Proof.* Note that

$$\begin{aligned}\|\overline{\mathbf{w}}_{t+1} - \mathbf{w}^*\|^2 &= \|\overline{\mathbf{w}}_{t+1} - \overline{\mathbf{v}}_{t+1} + \overline{\mathbf{v}}_{t+1} - \mathbf{w}^*\|^2 \\ &= \underbrace{\|\overline{\mathbf{w}}_{t+1} - \overline{\mathbf{v}}_{t+1}\|^2}_{A_1} + \underbrace{\|\overline{\mathbf{v}}_{t+1} - \mathbf{w}^*\|^2}_{A_2} + \underbrace{2\langle\overline{\mathbf{w}}_{t+1} - \overline{\mathbf{v}}_{t+1}, \overline{\mathbf{v}}_{t+1} - \mathbf{w}^*\rangle}_{A_3}.\end{aligned}$$

When expectation is taken over $\mathcal{S}_{t+1}$, the last term ($A_3$) vanishes due to the unbiasedness of $\overline{\mathbf{w}}_{t+1}$.

If $t + 1 \notin \mathcal{I}_E$, $A_1$ vanishes since $\overline{\mathbf{w}}_{t+1} = \overline{\mathbf{v}}_{t+1}$. We use Lemma 5 to bound $A_2$. Then it follows that

$$\mathbb{E}\|\overline{\mathbf{w}}_{t+1} - \mathbf{w}^*\|^2 \leq (1 - \eta_t\mu)\mathbb{E}\|\overline{\mathbf{w}}_t - \mathbf{w}^\star\|^2 + \eta_t^2 B.$$

If $t + 1 \in \mathcal{I}_E$, we additionally use Lemma 5 to bound $A_1$. Then

$$
\begin{aligned}
\mathbb{E}\left\|\overline{\mathbf{w}}_{t+1} - \mathbf{w}^*\right\|^2 &= \mathbb{E}\left\|\overline{\mathbf{w}}_{t+1} - \overline{\mathbf{v}}_{t+1}\right\|^2 + \mathbb{E}\left\|\overline{\mathbf{v}}_{t+1} - \mathbf{w}^*\right\|^2 \\
&\leq (1 - \eta_t \mu)\mathbb{E}\left\|\overline{\mathbf{w}}_t - \mathbf{w}^\star\right\|^2 + \eta_t^2 (B + C),
\end{aligned}
\tag{21}
$$

where $C$ is the upper bound of $\frac{1}{\eta_t^2}\mathbb{E}_{\mathcal{S}_t}\left\|\overline{\mathbf{v}}_{t+1} - \overline{\mathbf{w}}_{t+1}\right\|^2$ ($C$ is defined in Theorem 2 and 3).

The only difference between eqn. (21) and eqn. (11) is the additional $C$. Thus we can use the same argument there to prove the theorems here. Specifically, for a diminishing stepsize, $\eta_t = \frac{\beta}{t+\gamma}$ for some $\beta > \frac{1}{\mu}$ and $\gamma > 0$ such that $\eta_1 \leq \min\{\frac{1}{\mu}, \frac{1}{4L}\} = \frac{1}{4L}$ and $\eta_t \leq 2\eta_{t+E}$, we can prove $\mathbb{E}\left\|\overline{\mathbf{w}}_{t+1} - \mathbf{w}^*\right\|^2 \leq \frac{v}{\gamma+t}$ where $v = \max\left\{\frac{\beta^2(B+C)}{\beta\mu-1}, (\gamma+1)\|\mathbf{w}_1 - \mathbf{w}^*\|^2\right\}$.

Then by the strong convexity of $F(\cdot)$,

$$
\mathbb{E}[F(\overline{\mathbf{w}}_t)] - F^* \leq \frac{L}{2}\Delta_t \leq \frac{L}{2}\frac{v}{\gamma+t}.
$$

Specifically, if we choose $\beta = \frac{2}{\mu}, \gamma = \max\{8\frac{L}{\mu} - 1, E\}$ and denote $\kappa = \frac{L}{\mu}$, then $\eta_t = \frac{2}{\mu}\frac{1}{\gamma+t}$ and

$$
\mathbb{E}[F(\overline{\mathbf{w}}_t)] - F^* \leq \frac{2\kappa}{\gamma+t}\left(\frac{B+C}{\mu} + 2L\|\mathbf{w}_1 - \mathbf{w}^*\|^2\right).
$$

$\square$

### B.4 DEFERRED PROOFS OF KEY LEMMAS

*Proof of Lemma 4.* We first give a key observation which is useful to prove the followings. Let $\{x_i\}_{i=1}^N$ denote any fixed deterministic sequence. We sample a multiset $\mathcal{S}_t$ (with size $K$) by the procedure where for each sampling time, we sample $x_k$ with probability $q_k$ for each time. Pay attention that two samples are not necessarily independent. We only require each sampling distribution is identically. Let $\mathcal{S}_t = \{i_1, \cdots, i_K\} \subset [N]$ (some $i_k$'s may have the same value). Then

$$
\mathbb{E}_{\mathcal{S}_t}\sum_{k \in \mathcal{S}_t} x_k = \mathbb{E}_{\mathcal{S}_t}\sum_{k=1}^K x_{i_k} = K\mathbb{E}_{\mathcal{S}_t}x_{i_1} = K\sum_{k=1}^N q_k x_k.
$$

For Scheme I, $q_k = p_k$ and for Scheme II, $q_k = \frac{1}{N}$. It is easy to prove this lemma when equipped with this observation. $\square$

*Proof of Lemma 5.* We separately prove the bounded variance for two schemes. Let $\mathcal{S}_{t+1} = \{i_1, \cdots, i_K\}$ denote the multiset of chosen indexes.

(1) For Scheme I, $\overline{\mathbf{w}}_{t+1} = \frac{1}{K}\sum_{l=1}^K \mathbf{v}_{t+1}^{i_l}$. Taking expectation over $\mathcal{S}_{t+1}$, we have

$$
\mathbb{E}_{\mathcal{S}_t}\left\|\overline{\mathbf{w}}_{t+1} - \overline{\mathbf{v}}_{t+1}\right\|^2 = \mathbb{E}_{\mathcal{S}_t}\frac{1}{K^2}\sum_{l=1}^K\left\|\mathbf{v}_{t+1}^{i_l} - \overline{\mathbf{v}}_{t+1}\right\|^2 = \frac{1}{K}\sum_{k=1}^N p_k\left\|\mathbf{v}_{t+1}^k - \overline{\mathbf{v}}_{t+1}\right\|^2
\tag{22}
$$

where the first equality follows from $\mathbf{v}_{t+1}^{i_l}$ are independent and unbiased.

To bound eqn. (22), we use the same argument in Lemma 5. Since $t + 1 \in \mathcal{I}_E$, we know that the time $t_0 = t - E + 1 \in \mathcal{I}_E$ is the communication time, which implies $\{\mathbf{w}_{t_0}^k\}_{k=1}^N$ is identical. Then

$$
\begin{aligned}
\sum_{k=1}^N p_k\left\|\mathbf{v}_{t+1}^k - \overline{\mathbf{v}}_{t+1}\right\|^2 &= \sum_{k=1}^N p_k\left\|(\mathbf{v}_{t+1}^k - \overline{\mathbf{w}}_{t_0}) - (\overline{\mathbf{v}}_{t+1} - \overline{\mathbf{w}}_{t_0})\right\|^2 \\
&\leq \sum_{k=1}^N p_k\left\|\mathbf{v}_{t+1}^k - \overline{\mathbf{w}}_{t_0}\right\|^2
\end{aligned}
$$

where the last inequality results from $\sum_{k=1}^{N} p_k(\mathbf{v}_{t+1}^k - \overline{\mathbf{w}}_{t_0}) = \overline{\mathbf{v}}_{t+1} - \overline{\mathbf{w}}_{t_0}$ and $\mathbb{E}\|\mathbf{x} - \mathbb{E}\mathbf{x}\|^2 \leq \mathbb{E}\|\mathbf{x}\|^2$. Similarly, we have

$$
\begin{aligned}
\mathbb{E}_{\mathcal{S}_t}\left\|\overline{\mathbf{w}}_{t+1} - \overline{\mathbf{v}}_{t+1}\right\|^2 &\leq \frac{1}{K}\sum_{k=1}^{N} p_k \mathbb{E}\left\|\mathbf{v}_{t+1}^k - \overline{\mathbf{w}}_{t_0}\right\|^2 \\
&\leq \frac{1}{K}\sum_{k=1}^{N} p_k \mathbb{E}\left\|\mathbf{v}_{t+1}^k - \mathbf{w}_{t_0}^k\right\|^2 \\
&\leq \frac{1}{K}\sum_{k=1}^{N} p_k E \sum_{i=t_0}^{t} \mathbb{E}\left\|\eta_i \nabla F_k(\mathbf{w}_i^k, \xi_i^k)\right\|^2 \\
&\leq \frac{1}{K}E^2 \eta_{t_0}^2 G^2 \leq \frac{4}{K}\eta_t^2 E^2 G^2
\end{aligned}
$$

where in the last inequality we use the fact that $\eta_t$ is non-increasing and $\eta_{t_0} \leq 2\eta_t$.

(2) For Scheme II, when assuming $p_1 = p_2 = \cdots = p_N = \frac{1}{N}$, we again have $\overline{\mathbf{w}}_{t+1} = \frac{1}{K}\sum_{l=1}^{K} \mathbf{v}_{t+1}^{i_l}$.

$$
\begin{aligned}
\mathbb{E}_{\mathcal{S}_t}\left\|\overline{\mathbf{w}}_{t+1} - \overline{\mathbf{v}}_{t+1}\right\|^2 &= \mathbb{E}_{\mathcal{S}_t}\left\|\frac{1}{K}\sum_{i \in S_{t+1}} \mathbf{v}_{t+1}^i - \overline{\mathbf{v}}_{t+1}\right\|^2 = \frac{1}{K^2}\mathbb{E}_{\mathcal{S}_t}\left\|\sum_{i=1}^{N} \mathbb{I}\{i \in S_t\}(\mathbf{v}_{t+1}^i - \overline{\mathbf{v}}_{t+1})\right\|^2 \\
&= \frac{1}{K^2}\left[\sum_{i \in [N]} \mathbb{P}(i \in S_{t+1})\left\|\mathbf{v}_{t+1}^i - \overline{\mathbf{v}}_{t+1}\right\|^2 + \sum_{i \neq j} \mathbb{P}(i,j \in S_{t+1})\langle \mathbf{v}_{t+1}^i - \overline{\mathbf{v}}_{t+1}, \mathbf{v}_{t+1}^j - \overline{\mathbf{v}}_{t+1}\rangle\right] \\
&= \frac{1}{KN}\sum_{i=1}^{N}\left\|\mathbf{v}_{t+1}^i - \overline{\mathbf{v}}_{t+1}\right\|^2 + \sum_{i \neq j}\frac{K-1}{KN(N-1)}\langle \mathbf{v}_{t+1}^i - \overline{\mathbf{v}}_{t+1}, \mathbf{v}_{t+1}^j - \overline{\mathbf{v}}_{t+1}\rangle \\
&= \frac{1}{K(N-1)}\left(1 - \frac{K}{N}\right)\sum_{i=1}^{N}\left\|\mathbf{v}_{t+1}^i - \overline{\mathbf{v}}_{t+1}\right\|^2
\end{aligned}
$$

where we use the following equalities: (1) $\mathbb{P}(i \in S_{t+1}) = \frac{K}{N}$ and $\mathbb{P}(i,j \in S_{t+1}) = \frac{K(K-1)}{N(N-1)}$ for all $i \neq j$ and (2) $\sum_{i \in [N]}\left\|\mathbf{v}_{t+1}^i - \overline{\mathbf{v}}_{t+1}\right\|^2 + \sum_{i \neq j}\langle \mathbf{v}_{t+1}^i - \overline{\mathbf{v}}_{t+1}, \mathbf{v}_{t+1}^j - \overline{\mathbf{v}}_{t+1}\rangle = 0$.

Therefore,

$$
\begin{aligned}
\mathbb{E}\left\|\overline{\mathbf{w}}_{t+1} - \overline{\mathbf{v}}_{t+1}\right\|^2 &= \frac{N}{K(N-1)}\left(1 - \frac{K}{N}\right)\mathbb{E}\left[\frac{1}{N}\sum_{i=1}^{N}\left\|\mathbf{v}_{t+1}^i - \overline{\mathbf{v}}_{t+1}\right\|^2\right] \\
&\leq \frac{N}{K(N-1)}\left(1 - \frac{K}{N}\right)\mathbb{E}\left[\frac{1}{N}\sum_{i=1}^{N}\left\|\mathbf{v}_{t+1}^i - \overline{\mathbf{w}}_{t_0}\right\|^2\right] \\
&\leq \frac{N}{K(N-1)}\left(1 - \frac{K}{N}\right)4\eta_t^2 E^2 G^2.
\end{aligned}
$$

where in the last inequality we use the same argument in (1).  $\square$

## C   THE EMPIRICAL RISK MINIMIZATION EXAMPLE IN SECTION 4

### C.1   DETAIL OF THE EXAMPLE

Let $p > 1$ be a positive integer. To avoid the trivial case, we assume $N > 1$. Consider the following quadratic optimization

$$
\min_{\mathbf{w}} F(\mathbf{w}) \triangleq \frac{1}{2N}\left[\mathbf{w}^\top \mathbf{A}\mathbf{w} - 2\mathbf{b}^\top \mathbf{w}\right] + \frac{\mu}{2}\|\mathbf{w}\|_2^2, \tag{23}
$$

where $\mathbf{A} \in \mathbb{R}^{(Np+1)\times(Np+1)}$, $\mathbf{w}, \mathbf{b} \in \mathbb{R}^{Np+1}$ and $\mu > 0$. Specifically, let $\mathbf{b} = \mathbf{e}_1 \triangleq (1, 0, \cdots, 0)^\top$, and $\mathbf{A}$ be a symmetric and tri-diagonal matrix defined by

$$(\mathbf{A})_{i,j} = \begin{cases} 2, & i = j \in [1, Np+1], \\ -1, & |j - i| = 1 \text{ and } i, j \in [1, Np+1], \\ 0, & \text{otherwise,} \end{cases} \tag{24}$$

where $i, j$ are row and column indices, respectively. We partition $\mathbf{A}$ into a sum of $N$ symmetric matrices ($\mathbf{A} = \sum_{k=1}^{N} \mathbf{A}_k$) and $\mathbf{b}$ into $\mathbf{b} = \sum_{k=1}^{N} \mathbf{b}_k$. Specifically, we choose $\mathbf{b}_1 = \mathbf{b} = \mathbf{e}_1$ and $\mathbf{b}_2 = \cdots = \mathbf{b}_N = 0$. To give the formulation of $\mathbf{A}_k$'s, we first introduce a series of sparse and symmetric matrices $\mathbf{B}_k$ ($1 \le k \le N$):

$$(\mathbf{B}_k)_{i,j} = \begin{cases} 1, & i = j \in \{(k-1)p+1, kp+1\}, \\ 2, & i = j \text{ and } (k-1)p+1 < i, j < kp+1, \\ -1, & |j - i| = 1 \text{ and } i, j \in [(k-1)p+1, kp+1], \\ 0, & \text{otherwise.} \end{cases} \tag{25}$$

Now $\mathbf{A}_k$'s are given by $\mathbf{A}_1 = \mathbf{B}_1 + \mathbf{E}_{1,1}$, $\mathbf{A}_k = \mathbf{B}_k$ ($2 \le k \le N-1$) and $\mathbf{A}_N = \mathbf{B}_N + \mathbf{E}_{Np+1,Np+1}$, where $\mathbf{E}_{i,j}$ is the matrix where only the $(i, j)$th entry is one and the rest are zero.

Back to the federated setting, we distribute the $k$-th partition $(\mathbf{A}_k, \mathbf{b}_k)$ to the $k$-th device and construct its corresponding local objective by

$$F_k(\mathbf{w}) \triangleq \frac{1}{2} \left[ \mathbf{w}^\top \mathbf{A}_k \mathbf{w} - 2\mathbf{b}_k^\top \mathbf{w} + \mu \|\mathbf{w}\|_2^2 \right]. \tag{26}$$

In the next subsection (Appendix C.3), we show that the quadratic minimization with the global objective (23) and the local objectives (26) is actually a distributed linear regression. In this example, training data are not identically but balanced distributed. Moreover, data in each device are sparse in the sense that non-zero features only occur in one block. The following theorem (Theorem 5) shows that `FedAvg` might converge to sub-optimal points even if the learning rate is small enough. We provide a numerical illustration in Appendix C.2 and a mathematical proof in Appendix C.4.

**Theorem 5.** *In the above problem of the distributed linear regression, assume that each device computes exact gradients (which are not stochastic). With a constant and small enough learning rate $\eta$ and $E > 1$, `FedAvg` converges to a sub-optimal solution, whereas `FedAvg` with $E = 1$ (i.e., gradient descent) converges to the optimum. Specifically, in a quantitative way, we have*

$$\|\widetilde{\mathbf{w}}^* - \mathbf{w}^*\| \ge \frac{(E-1)\eta}{16} \|\mathbf{A}_1 \mathbf{A}_2 \mathbf{w}^*\|$$

*where $\widetilde{\mathbf{w}}^*$ is the solution produced by `FedAvg` and $\mathbf{w}^*$ is the optimal solution.*

## C.2 NUMERICAL ILLUSTRATION ON THE EXAMPLE

We conduct a few numerical experiments to illustrate the poor performance of `FedAvg` on the example introduced in Section 4. Here we set $N = 5, p = 4, \mu = 2 \times 10^{-4}$. The annealing scheme of learning rates is given by $\eta_t = \frac{1/5}{5 + t \cdot a}$ where $a$ is the best parameter chosen from the set $\{10^{-2}, 10^{-4}, 10^{-6}\}$.

## C.3 SOME PROPERTIES OF THE EXAMPLE

Recall that the symmetric matrix $\mathbf{A} \in \mathbb{R}^{(Np+1)\times(Np+1)}$ is defined in eqn. (24). Observe that $\mathbf{A}$ is invertible and for all vector $\mathbf{w} \in \mathbb{R}^{Np+1}$,

$$\mathbf{w}^\top \mathbf{A} \mathbf{w} = 2 \sum_{i=1}^{Np+1} \mathbf{w}_i^2 - 2 \sum_{i=1}^{Np} \mathbf{w}_i \mathbf{w}_{i+1} = \mathbf{w}_1^2 + \mathbf{w}_{Np+1}^2 + \sum_{i=1}^{Np} (\mathbf{w}_i - \mathbf{w}_{i+1})^2 \le 4\|\mathbf{w}\|_2^2. \tag{27}$$

which implies that $0 \prec \mathbf{A} \preceq 4\mathbf{I}$.

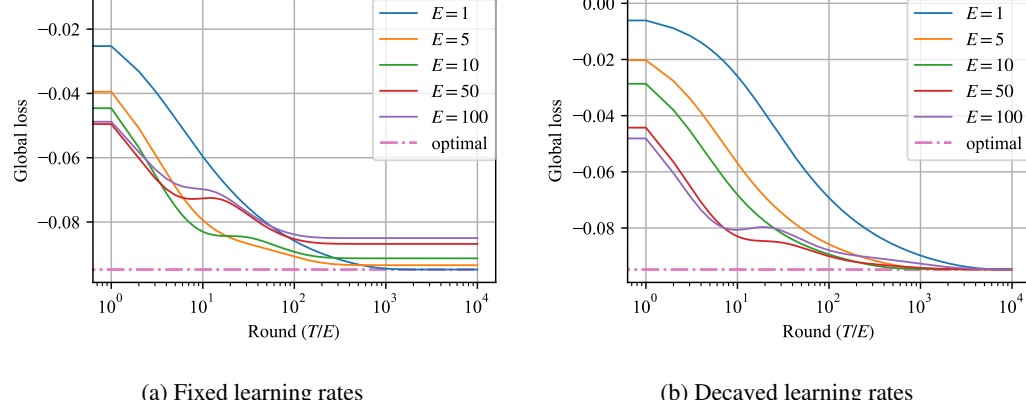

(a) Fixed learning rates                    (b) Decayed learning rates

Figure 2: The left figure shows that the global objective value that `FedAvg` converges to is not optimal unless $E = 1$. Once we decay the learning rate, `FedAvg` can converge to the optimal even if $E > 1$.

The sparse and symmetric matrices $\mathbf{B}_k$ $(1 \le k \le N)$ defined in eqn. (25) can be rewritten as

$$
(\mathbf{B}_k) = \begin{pmatrix} \mathbf{0}_{(k-1)p \times (k-1)p} & & \\ & \begin{pmatrix} 1 & -1 & & & \\ -1 & 2 & -1 & & \\ & -1 & \ddots & \ddots & \\ & & \ddots & 2 & -1 \\ & & & -1 & 1 \end{pmatrix}_{(p+1) \times (p+1)} & \\ & & \mathbf{0}_{(N-k)p \times (N-k)p} \end{pmatrix}.
$$

From theory of linear algebra, it is easy to follow this proposition.

**Proposition 1.** *By the way of construction, $\mathbf{A}_k$'s have following properties:*

1. *$\mathbf{A}_k$ is positive semidefinite with $\|\mathbf{A}_k\|_2 \le 4$;*

2. *$\mathrm{rank}(\mathbf{A}_2) = \cdots = \mathrm{rank}(\mathbf{A}_{N-1}) = p$ and $\mathrm{rank}(\mathbf{A}_1) = \mathrm{rank}(\mathbf{A}_N) = p + 1$;*

3. *For each $k$, there exist a matrix $\mathbf{X}_k \in \mathbb{R}^{r_k \times (Np+1)}$ such that $\mathbf{A}_k = \mathbf{X}_k^\top \mathbf{X}_k$ where $r_k = \mathrm{rank}(\mathbf{A}_k)$. Given any $k$, each row of $\mathbf{X}_k$ has non-zero entries only on a block of coordinates, namely $\mathcal{I}_k = \{(k-1)p+1, (k-1)p+2, \cdots, kp+1\}$. As a result, $\mathbf{A} = \sum_{k=1}^{N} \mathbf{A}_k = \mathbf{X}^\top \mathbf{X}$, where $\mathbf{X} = (\mathbf{X}_1^\top, \cdots, \mathbf{X}_N^\top)^\top \in \mathbb{R}^{(Np+2) \times (Np+1)}$.*

4. *$\mathbf{w}^* = \mathbf{A}^{-1} \mathbf{b}$ is the global minimizer of problem eqn. (23), given by $(\mathbf{w}^*)_i = 1 - \frac{i}{Np+2}$ $(1 \le i \le Np+1)$. Let $\widetilde{\mathbf{w}} \triangleq (\underbrace{1, \cdots, 1}_{p+1}, \underbrace{0, \cdots, 0}_{(N-1)p})^\top \in \mathbb{R}^{Np+1}$, then $\mathbf{A}_1 \widetilde{\mathbf{w}} = \mathbf{X}_1^\top \mathbf{X}_1 \widetilde{\mathbf{w}} = \mathbf{b}_1$.*

From Proposition 1, we can rewrite these local quadratic objectives in form of a ridge linear regression. Specifically, for $k = 1$,

$$
\begin{aligned}
F_1(\mathbf{w}) &= \frac{1}{2} \left[ \mathbf{w}^\top \mathbf{A}_1 \mathbf{w} - 2\mathbf{b}_1^\top \mathbf{w} + \mu \|\mathbf{w}\|^2 \right], \\
&= \frac{1}{2} \left[ \mathbf{w}^\top \mathbf{X}_1^\top \mathbf{X}_1 \mathbf{w} - 2\widetilde{\mathbf{w}}^\top \mathbf{X}_1^\top \mathbf{X}_1 \mathbf{w} + \mu \|\mathbf{w}\|^2 \right], \\
&= \frac{1}{2} \|\mathbf{X}_1 (\mathbf{w} - \widetilde{\mathbf{w}})\|_2^2 + \frac{1}{2} \mu \|\mathbf{w}\|^2 + C,
\end{aligned}
$$

where $C$ is some constant irrelevant with $\mathbf{w}$). For $2 \leq k \leq N$,

$$F_k(\mathbf{w}) = \frac{1}{2} \left[ \mathbf{w}^\top \mathbf{A}_k \mathbf{w} - 2\mathbf{b}_k^\top \mathbf{w} + \mu \|\mathbf{w}\|^2 \right],$$

$$= \frac{1}{2} \|\mathbf{X}_k \mathbf{w}\|_2^2 + \frac{1}{2} \mu \|\mathbf{w}\|^2.$$

Similarly, the global quadratic objective eqn. (23) can be written as $F(\mathbf{w}) = \frac{1}{2N} \|\mathbf{X}(\mathbf{w} - \mathbf{w}^*)\|_2^2 + \frac{1}{2}\mu \|\mathbf{w}\|^2$.

Data in each device are sparse in the sense that non-zero features only occur in the block $\mathcal{I}_k$ of coordinates. Blocks on neighboring devices only overlap one coordinate, i.e., $|\mathcal{I}_k \cap \mathcal{I}_{k+1}| = 1$. These observations imply that the training data in this example is not identically distributed.

The $k$-th device has $r_k$ $(= p$ or $p+1)$ non-zero feature vectors which are vertically concatenated into the feature matrix $\mathbf{X}_k$. Without loss of generality, we can assume all devices hold $p+1$ data points since we can always add additional zero vectors to expand the local dataset. Therefore $n_1 = \cdots = n_N = p+1$ in this case, which implies that the training data in this example is balanced distributed.

## C.4    PROOF OF THEOREM 5.

*Proof of Theorem 5.* To prove the theorem, we assume that (i) all devices hold the same amount of data points, (ii) all devices perform local updates in parallel, (iii) all workers use the same learning rate $\eta$ and (iv) all gradients computed by each device make use of its full local dataset (hence this case is a deterministic optimization problem). We first provide the result when $\mu = 0$.

For convenience, we slightly abuse the notation such that $\mathbf{w}_t$ is the global parameter at round $t$ rather than step $t$. Let $\mathbf{w}_t^{(k)}$ the updated local parameter at $k$-th worker at round $t$. Once the first worker that holds data $(\mathbf{A}_1, \mathbf{b}_1)$ runs $E$ step of SGD on $F_1(\mathbf{w})$ from $\mathbf{w}_t$, it follows that

$$\mathbf{w}_t^{(1)} = (\mathbf{I} - \eta \mathbf{A}_1)^E \mathbf{w}_t + \eta \sum_{l=0}^{E-1} (\mathbf{I} - \eta \mathbf{A}_1)^l \mathbf{b}_1.$$

For the rest of workers, we have $\mathbf{w}_t^{(k)} = (\mathbf{I} - \eta \mathbf{A}_i)^E \mathbf{w}_t$ $(2 \leq k \leq N)$.

Therefore, from the algorithm,

$$\mathbf{w}_{t+1} = \frac{1}{N} \sum_{k=1}^{N} \mathbf{w}_{t+1}^{(k)} = \left( \frac{1}{N} \sum_{i=1}^{N} (\mathbf{I} - \eta \mathbf{A}_i)^E \right) \mathbf{w}_t + \frac{\eta}{N} \sum_{l=0}^{E-1} (\mathbf{I} - \eta \mathbf{A}_1)^l \mathbf{b}_1.$$

Define $\rho \triangleq \|\frac{1}{N} \sum_{i=1}^{N} (\mathbf{I} - \eta \mathbf{A}_i)^E\|_2$. Next we show that when $\eta < \frac{1}{4}$, we have $\rho < 1$. From Proposition 1, $\|\mathbf{A}_k\|_2 \leq 4$ and $\mathbf{A}_k \succeq 0$ for $\forall\, k \in [N]$. This means $\|\mathbf{I} - \eta \mathbf{A}_k\|_2 \leq 1$ for all $k \in [N]$. Then for any $\mathbf{x} \in \mathbb{R}^{Np+1}$ and $\|x\|_2 = 1$, we have $\mathbf{x}^\top (\mathbf{I} - \eta \mathbf{A}_k)^E \mathbf{x} \leq 1$ and it is monotonically decreasing when $E$ is increasing. Then

$$\mathbf{x}^\top \left( \frac{1}{N} \sum_{i=1}^{N} (\mathbf{I} - \eta \mathbf{A}_i)^E \right) \mathbf{x} \leq \mathbf{x}^\top \left( \frac{1}{N} \sum_{i=1}^{N} (\mathbf{I} - \eta \mathbf{A}_i) \right) \mathbf{x}$$

$$= \mathbf{x}^\top \left( \mathbf{I} - \frac{\eta}{N} \mathbf{A} \right) \mathbf{x} < 1$$

since $0 \prec \mathbf{A} \preceq 4\mathbf{I}$ means $0 \preceq (\mathbf{I} - \frac{\eta}{N} \mathbf{A}) \prec \mathbf{I}$.

Then $\|\mathbf{w}_{t+1} - \mathbf{w}_t\|_2 \leq \rho \|\mathbf{w}_t - \mathbf{w}_{t-1}\|_2 \leq \rho^t \|\mathbf{w}_1 - \mathbf{w}_0\|_2$. By the triangle inequality,

$$\|\mathbf{w}_{t+n} - \mathbf{w}_t\|_2 \leq \sum_{i=0}^{n-1} \|\mathbf{w}_{t+i+1} - \mathbf{w}_{t+i}\|_2 \leq \sum_{i=0}^{n-1} \rho^{t+i} \|\mathbf{w}_1 - \mathbf{w}_0\|_2 \leq \rho^t \frac{\|\mathbf{w}_1 - \mathbf{w}_0\|_2}{1 - \rho}$$

which implies that $\{\mathbf{w}_t\}_{t \geq 1}$ is a Cauchy sequence and thus has a limit denoted by $\widetilde{\mathbf{w}}^*$. We have

$$\widetilde{\mathbf{w}}^* = \left( \mathbf{I} - \frac{1}{N} \sum_{i=1}^{N} (\mathbf{I} - \eta \mathbf{A}_i)^E \right)^{-1} \left[ \frac{\eta}{N} \sum_{l=0}^{E-1} (\mathbf{I} - \eta \mathbf{A}_1)^l \mathbf{b} \right]. \tag{28}$$

Now we can discuss the impact of $E$.

(1) When $E = 1$, it follows from eqn. (28) that $\widetilde{\mathbf{w}}^* = \mathbf{A}^{-1}\mathbf{b} = \mathbf{w}^*$, i.e., `FedAvg` converges to the global minimizer.

(2) When $E = \infty$, $\lim\limits_{E \to \infty} \eta \sum_{l=0}^{E-1} (\mathbf{I} - \eta\mathbf{A}_1)^l \mathbf{b} = \mathbf{A}_1^+ \mathbf{b}_1 = \widetilde{\mathbf{w}}$ and $\lim\limits_{E \to \infty} \frac{1}{N} \sum_{i=1}^{N} (\mathbf{I} - \eta\mathbf{A}_i)^E =$ $\mathrm{diag}\{(1 - \frac{1}{N})\mathbf{I}_p; (1 - \frac{1}{N})\mathbf{I} - \frac{1}{N}\mathbf{M}; (1 - \frac{1}{N})\mathbf{I}_p\}$ where $\mathbf{M} \in \mathbb{R}^{(N-2)p+1 \times (N-2)p+1}$ is some a symmetric matrix. Actually $\mathbf{M}$ is **almost** a diagonal matrix in the sense that there are totally $N-2$ completely the same matrices (i.e., $\frac{1}{p+1}\mathbf{e}\mathbf{e}^T \in \mathbb{R}^{(p+1) \times (p+1)}$) placed on the diagonal of $\mathbf{M}$ but each overlapping only the lower right corner element with the top left corner element of the next block. Therefore $\widetilde{\mathbf{w}}^* = (\underbrace{1, \cdots, 1}_{p}, \underbrace{\mathbf{V}_{11}, \cdots, \mathbf{V}_{(N-2)p+1,1}}_{(N-2)p+1}, \underbrace{0, \cdots, 0}_{p})^T$ where $\mathbf{V} = (\mathbf{I} - \mathbf{M})^{-1}$. From (4) of Proposition 1, $\widetilde{\mathbf{w}}^*$ is different from $\mathbf{w}^*$

(3) When $2 \le E < \infty$, note that
$$\widetilde{\mathbf{w}}^* - \mathbf{w}^*$$
$$= \left(\mathbf{I} - \frac{1}{N}\sum_{i=1}^{N}(\mathbf{I} - \eta\mathbf{A}_i)^E\right)^{-1} \left[\frac{\eta}{N}\sum_{l=0}^{E-1}(\mathbf{I} - \eta\mathbf{A}_1)^l\mathbf{A} - \left(\mathbf{I} - \frac{1}{N}\sum_{i=1}^{N}(\mathbf{I} - \eta\mathbf{A}_i)^E\right)\right]\mathbf{w}^*. \tag{29}$$

The right hand side of the last equation cannot be zero. Quantificationally speaking, we have the following lemma. We defer the proof for the next subsection.

**Lemma 6.** *If the step size $\eta$ is sufficiently small, then in this example, we have*
$$\|\widetilde{\mathbf{w}}^* - \mathbf{w}^*\| \ge \frac{(E - 1)\eta}{16}\|\mathbf{A}_1\mathbf{A}_2\mathbf{w}^*\|. \tag{30}$$

Since $\mathbf{A}_1\mathbf{A}_2 \ne \mathbf{0}$ and $\mathbf{w}^*$ is dense, the lower bound in eqn. (30) is not vacuous.

Now we have proved the result when $\mu = 0$. For the case where $\mu > 0$, we replace $\mathbf{A}_i$ with $\mathbf{A}_i + \mu\mathbf{I}$ and assume $\mu < \frac{1}{4+\mu}$ instead of the original. The discussion on different choice of $E$ is unaffected. $\square$

## C.5 PROOF OF LEMMA 6

*Proof.* We will derive the conclusion mainly from the expression eqn. (29). Let $f(\eta)$ be a function of $\eta$. We say a matrix $\mathbf{T}$ is $\Theta(f(\eta))$ if and only if there exist some positive constants namely $C_1$ and $C_2$ such that $C_1 f(\eta) \le \|\mathbf{T}\| \le C_2 f(\eta)$ for all $\eta > 0$. In the following analysis, we all consider the regime where $\eta$ is sufficiently small.

Denote by $\mathbf{V} = \sum_{i=1}^{N} \mathbf{A}_i^2$. First we have
$$\mathbf{I} - \frac{1}{N}\sum_{i=1}^{N}(\mathbf{I} - \eta\mathbf{A}_i)^E = \mathbf{I} - \frac{1}{N}\sum_{i=1}^{N}\left(\mathbf{I} - E\eta\mathbf{A}_i + \frac{E(E-1)}{2}\eta^2\mathbf{A}_i^2 + \Theta(\eta^3)\right)$$
$$= \frac{E\eta}{N}\mathbf{A} - \frac{E(E-1)}{2N}\eta^2\mathbf{V} + \Theta(\eta^3). \tag{31}$$

Then by plugging this equation into the right hand part of eqn. (29), we have
$$\frac{\eta}{N}\sum_{l=0}^{E-1}(\mathbf{I} - \eta\mathbf{A}_1)^l\mathbf{A} - \left(\mathbf{I} - \frac{1}{N}\sum_{i=1}^{N}(\mathbf{I} - \eta\mathbf{A}_i)^E\right)$$
$$= \frac{\eta}{N}\sum_{l=0}^{E-1}(\mathbf{I} - l\eta\mathbf{A}_1 + \Theta(\eta^2))\mathbf{A} - \left(\frac{E\eta}{N}\mathbf{A} - \frac{E(E-1)}{2N}\eta^2\mathbf{V} + \Theta(\eta^3)\right)$$
$$= \frac{\eta^2}{N}\left(\frac{E(E-1)}{2}(\mathbf{V} - \mathbf{A}_1\mathbf{A}) + \Theta(\eta)\right)$$

Second from eqn. (31), we have that

$$\left(\mathbf{I} - \frac{1}{N}\sum_{i=1}^{N}(\mathbf{I} - \eta\mathbf{A}_i)^E\right)^{-1} = \left(\frac{E\eta}{N}\mathbf{A} + \Theta(\eta^2)\right)^{-1} = \frac{N}{E\eta}\mathbf{A}^{-1} + \Theta(1).$$

Plugging the last two equations into eqn. (29), we have

$$
\begin{aligned}
\|\widetilde{\mathbf{w}}^* - \mathbf{w}^*\| &= \left\|\left(\frac{N}{E\eta}\mathbf{A}^{-1} + \Theta(1)\right)\frac{\eta^2}{N}\left(\frac{E(E-1)}{2}(\mathbf{V} - \mathbf{A}_1\mathbf{A}) + \Theta(\eta)\right)\mathbf{w}^*\right\| \\
&= \left\|\left(\frac{E-1}{2}\eta\mathbf{A}^{-1}(\mathbf{V} - \mathbf{A}_1\mathbf{A}) + \Theta(\eta)\right)\mathbf{w}^*\right\| \\
&\geq \frac{(E-1)\eta}{16}\|(\mathbf{V} - \mathbf{A}_1\mathbf{A})\mathbf{w}^*\| \\
&= \frac{(E-1)\eta}{16}\|\mathbf{A}_1\mathbf{A}_2\mathbf{w}^*\|
\end{aligned}
$$

where the last inequality holds because (i) we require $\eta$ to be sufficiently small and (ii) $\|\mathbf{A}^{-1}\mathbf{x}\| \geq \frac{1}{4}\|\mathbf{x}\|$ for any vector $\mathbf{x}$ as a result of $0 < \|\mathbf{A}\| \leq 4$. The last equality uses the fact (i) $\mathbf{V} - \mathbf{A}_1\mathbf{A} = \mathbf{A}_1\sum_{i=2}^{n}\mathbf{A}_i$ and (ii) $\mathbf{A}_1\mathbf{A}_i = \mathbf{0}$ for any $i \geq 3$. $\square$

## D EXPERIMENTAL DETAILS

### D.1 EXPERIMENTAL SETTING

**Model and loss.** We examine our theoretical results on a multinomial logistic regression. Specifically, let $f(\mathbf{w}; x_i)$ denote the prediction model with the parameter $\mathbf{w} = (\mathbf{W}, \mathbf{b})$ and the form $f(\mathbf{w}; \mathbf{x}_i) = \text{softmax}(\mathbf{W}\mathbf{x}_i + \mathbf{b})$. The loss function is given by

$$F(\mathbf{w}) = \frac{1}{n}\sum_{i=1}^{n}\text{CrossEntropy}\left(f(\mathbf{w}; \mathbf{x}_i), \mathbf{y}_i\right) + \lambda\|\mathbf{w}\|_2^2.$$

This is a convex optimization problem. The regularization parameter is set to $\lambda = 10^{-4}$.

**Datasets.** We evaluate our theoretical results on both real data and synthetic data. For real data, we choose MNIST dataset (LeCun et al., 1998) because of its wide academic use. To impose statistical heterogeneity, we distribute the data among $N = 100$ devices such that each device contains samples of only two digits. To explore the effect of data unbalance, we further vary the number of samples among devices. Specifically, for unbalanced cases, the number of samples among devices follows a power law, while for balanced cases, we force all devices to have the same amount of samples.

Synthetic data allow us to manipulate heterogeneity more precisely. Here we follow the same setup as described in (Sahu et al., 2018). In particular, we generate synthetic samples $(\mathbf{X}_k, \mathbf{Y}_k)$ according to the model $y = \text{argmax}(\text{softmax}(\mathbf{W}_k x + \mathbf{b}_k))$ with $x \in \mathbb{R}^{60}, \mathbf{W}_k \in \mathbb{R}^{10\times 60}$ and $\mathbf{b}_k \in \mathbb{R}^{10}$, where $\mathbf{X}_k \in \mathbb{R}^{n_k \times 60}$ and $\mathbf{Y}_k \in \mathbb{R}^{n_k}$. We model each entry of $\mathbf{W}_k$ and $\mathbf{b}_k$ as $\mathcal{N}(\mu_k, 1)$ with $\mu_k \sim \mathcal{N}(0, \alpha)$, and $(x_k)_j \sim \mathcal{N}(v_k, \frac{1}{j^{1.2}})$ with $v_k \sim \mathcal{N}(B_k, 1)$ and $B_k \sim \mathcal{N}(0, \beta)$. Here $\alpha$ and $\beta$ allow for more precise manipulation of data heterogeneity: $\alpha$ controls how much local models differ from each other and $\beta$ controls how much the local data at each device differs from that of other devices. There are $N = 100$ devices in total. The number of samples $n_k$ in each device follows a power law, i.e., data are distributed in an unbalanced way. We denote by synthetic$(\alpha, \beta)$ the synthetic dataset with parameter $\alpha$ and $\beta$.

We summarize the information of federated datasets in Table 2.

**Experiments.** For all experiments, we initialize all runnings with $\mathbf{w}_0 = 0$. In each round, all selected devices run $E$ steps of SGD in parallel. We decay the learning rate at the end of each round by the following scheme $\eta_t = \frac{\eta_0}{1+t}$, where $\eta_0$ is chosen from the set $\{1, 0.1, 0.01\}$. We evaluate the averaged model after each global synchronization on the corresponding global objective. For fair comparison, we control all randomness in experiments so that the set of activated devices is the same across all different algorithms on one configuration.

Table 2: Statistics of federated datasets

| Dataset | Details | # Devices ($N$) | #Training samples ($n$) | Samples/device | |
|---|---|---|---|---|---|
| | | | | mean | std |
| MNIST | balanced | 100 | 54200 | 542 | 0 |
| | unbalanced | 100 | 62864 | 628 | 800 |
| Synthetic Data | $\alpha = 0, \beta = 0$ | 100 | 42522 | 425 | 1372 |
| | $\alpha = 1, \beta = 1$ | 100 | 27348 | 273 | 421 |

### D.2 THEORETICAL VERIFICATION

**The impact of $E$.** From our theory, when the total steps $T$ is sufficiently large, the required number of communication rounds to achieve a certain precision is

$$T_\epsilon / E \approx \mathcal{O}\left( \frac{EG^2}{K} + EG^2 + \frac{\sum_{k=1}^{N} p_k^2 \sigma^2 + L\Gamma + \kappa G^2}{E} \right),$$

which is s a function of $E$ that first decreases and then increases. This implies that the optimal local step $E^*$ exists. What's more, the $T_\epsilon / E$ evaluated at $E^*$ is

$$\mathcal{O}\left( G \sqrt{\sum_{k=1}^{N} p_k^2 \sigma^2 + L\Gamma + \kappa G^2} \right),$$

which implies that `FedAvg` needs more communication rounds to tackle with severer heterogeneity.

To validate these observations, we test `FedAvg` with Scheme I on our four datasets as listed in Table 2. In each round, we activate $K = 30$ devices and set $\eta_0 = 0.1$ for all experiments in this part. For unbalanced MNIST, we use batch size $b = 64$. The target loss value is $0.29$ and the minimum loss value found is $0.2591$. For balanced MNIST, we also use batch size $b = 64$. The target loss value is $0.50$ and the minimum loss value found is $0.3429$. For two synthetic datasets, we choose $b = 24$. The target loss value for synthetic(0,0) is $0.95$ and the minimum loss value is $0.7999$. Those for synthetic(1,1) are $1.15$ and $1.075$.

**The impact of $K$.** Our theory suggests that a larger $K$ may accelerate convergence since $T_\epsilon / E$ contains a term $\mathcal{O}\left( \frac{EG^2}{K} \right)$. We fix $E = 5$ and $\eta_0 = 0.1$ for all experiments in this part. We set the batch size to 64 for two MNIST datasets and 24 for two synthetic datasets. We test Scheme I for illustration. Our results show that, no matter what value $K$ is, `FedAvg` converges. From Figure 3, all the curves in each subfigure overlap a lot. To show more clearly the differences between the curves, we zoom in the last few rounds in the upper left corner of the figure. It reveals that the curve of a large enough $K$ is slightly better. This result also shows that there is no need to sample as many devices as possible in convex federated optimization.

**Sampling and averaging schemes.** We analyze the influence of sampling and averaging schemes. As stated in Section 3.3, **Scheme I** iid samples (with replacement) $K$ indices with weights $p_k$ and simply averages the models, which is proposed by Sahu et al. (2018). **Scheme II** uniformly samples (without replacement) $K$ devices and weightedly averages the models with scaling factor $N/K$. **Transformed Scheme II** scales each local objective and uses uniform sampling and simple averaging. We compare Scheme I, Scheme II and transformed Scheme II, as well as the original scheme (McMahan et al., 2017) on four datasets. We carefully tuned the learning rate for the original scheme. In particular, we choose the best step size from the set $\{0.1, 0.5, 0.9, 1.1\}$. We did not fine tune the rest schemes and set $\eta_0 = 0.1$ by default. The hyperparameters are the same for all schemes: $E = 20, K = 10$ and $b = 64$. The results are shown in Figure 1c and 1d.

Our theory renders Scheme I the guarantee of convergence in common federated setting. As expected, Scheme I performs well and stably across most experiments. This also coincides with the findings of Sahu et al. (2018). They noticed that Scheme I performs slightly better than another scheme: server

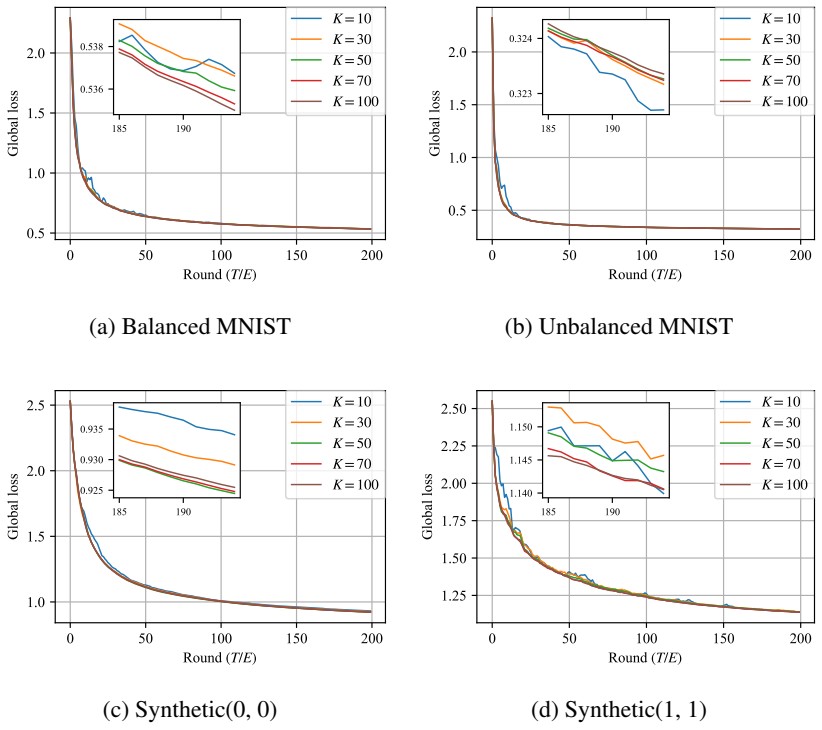

(a) Balanced MNIST    (b) Unbalanced MNIST

(c) Synthetic(0, 0)    (d) Synthetic(1, 1)

Figure 3: The impact of $K$ on four datasets. To show more clearly the differences between the curves, we zoom in the last few rounds in the upper left corner of the box.

first uniformly samples devices and then averages local models with weight $p_k / \sum_{l \in S_t} p_l$. However, our theoretical framework cannot apply to it, since for $t \in \mathcal{I}$, $\mathbb{E}_{S_t} \overline{\mathbf{w}}_t = \overline{\mathbf{v}}_t$ does not hold in general.

Our theory **does not** guarantee FedAvg with Scheme II could converge when the training data are unbalanced distributed. Actually, if the number of training samples varies too much among devices, Scheme II may even diverge. To illustrate this point, we have shown the terrible performance on mnist unbalanced dataset in Figure 1b. In Figure 4, we show additional results of Scheme II on the two synthetic datasets, which are the most unbalanced. We choose $b = 24$, $K = 10$, $E = 10$ and $\eta_0 = 0.1$ for these experiments. However, transformed Scheme II performs well except that it has a lower convergence rate than Scheme I.

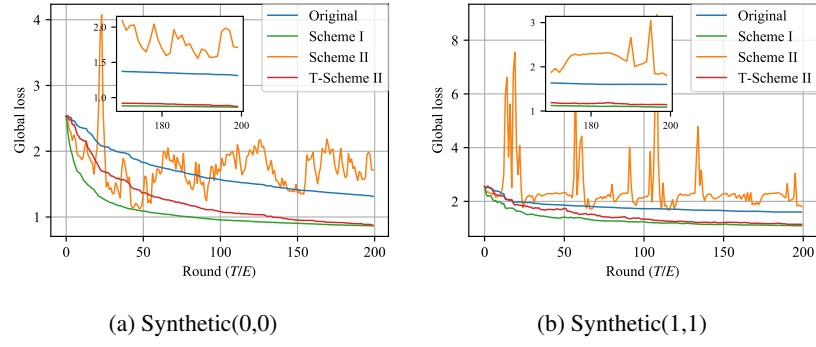

(a) Synthetic(0,0)    (b) Synthetic(1,1)

Figure 4: The performance of four schemes on two synthetic datasets. The Scheme I performs stably and the best. The original performs the second. The curve of the Scheme II fluctuates and has no sign of convergence. Transformed Scheme II has a lower convergence rate than Scheme I.

