# OpenReview forum: "On the Convergence of FedAvg on Non-IID Data"
_ICLR.cc/2020/Conference — Accept (Talk)_

### Official Review · AnonReviewer4 · 2019-10-22
**Official Blind Review #4**

**Rating:** 8

**Review:**

This paper presents convergence rates for straggler-aware averaged SGD for non-identically but independent distributed data. The paper is well-written and motivated with good discussions of the algorithm and the related works. The proof techniques involve bounding how much worse can the algorithm do because of non-identical distribution and introduction of stragglers into the standard analysis of SGD-like algorithms. The presented theory is useful, and also provides new insights such as a new sampling scheme and an inherent bias for the case of non-decaying step size. The empirical evaluation is adequate and well-presented. I think this paper is a strong contribution and should spark further discussions in the community.

**Experience Assessment:**

I have published one or two papers in this area.

**Review Assessment: Checking Correctness Of Derivations And Theory:**

I assessed the sensibility of the derivations and theory.

**Review Assessment: Checking Correctness Of Experiments:**

I carefully checked the experiments.

**Review Assessment: Thoroughness In Paper Reading:**

I read the paper at least twice and used my best judgement in assessing the paper.

---

> ### Author Response · Authors · 2019-11-07
> **Thank you for the supportive review**
>
> We greatly appreciate the reviewer's effort. Thanks for your positive reviews.

---

### Official Review · AnonReviewer1 · 2019-10-23
**Official Blind Review #1**

**Rating:** 8

**Review:**

This paper analyzes the convergence of FedAvg, the most popular algorithm for federated learning. The highlight of the paper is removing the following two assumptions: (i) the data are iid across devices, and (ii) all the devices are active. For smooth and strongly convex problems, the paper proves an O(1/T) convergence rate to global optimum for learning rate decaying like 1/t with time. It is also shown that with constant learning rate eta, the solution found can be necessarily Omega(eta) away from the optimum (for a specific problem instance), thus justifying the decaying learning rate used in the positive result.

Federated learning has been an important and popular research area since it models a highly distributed and heterogeneous learning system in real world. Previous theoretical analysis of FedAvg was quite scarce and either made the iid data assumption or required averaging all the devices. This work is the first to prove a convergence guarantee without these two assumptions. In particular, it only requires averaging a (random) subset of devices each round, which is much more realistic than averaging all.

I don't quite have an intuition for why you need strong convexity. I hope the authors could explain this in words and maybe comment on what are the challenges of removing this assumption.


------
Thanks to the authors for their response.

**Experience Assessment:**

I do not know much about this area.

**Review Assessment: Checking Correctness Of Derivations And Theory:**

I did not assess the derivations or theory.

**Review Assessment: Checking Correctness Of Experiments:**

I assessed the sensibility of the experiments.

**Review Assessment: Thoroughness In Paper Reading:**

I made a quick assessment of this paper.

---

> ### Author Response · Authors · 2019-11-07
> **Thank you for your supportive review**
>
> We greatly appreciate the reviewer's effort. Here are our responses to your comments.
>
> There are several benefits when we assume strong convexity.
> First, it facilitates our analysis.
> We have more ways to prove the convergence since there is no difference when we prove convergence for $\|w_t - w^*\|$ or $f(w_t) - f(w^*)$.
> Second, strong convexity and smoothness mean fast convergence rate $O(1/T)$.
> Since FedAvg is a distributed variant of SGD, it couldn't achieve faster convergence than SGD.
> We can remove the strong convexity assumption but the convergence rate is $O(1/\sqrt{T})$ (see Khaled et al. (2019)).
> Third, in Theorems 1 and 2, we require the learning rate $\eta_t = 2/(\gamma + t) 1/\mu$, which makes the use of a positive $\mu$.

---

### Official Review · AnonReviewer3 · 2019-10-24
**Official Blind Review #3**

**Rating:** 6

**Review:**

Federated learning is distinguished from the standard distributed learning in the following sense:
1) training is distributed over a huge number (say N) of devices and communication between the central server and devices are slow.
2) The central server has no control of individual devices, and there are inactive devices that does not respond to the server; full participation of all devices is unrealistic.
3) The local data distribution at each device is different from each other; i.e., the data is non-iid.

Due to property 1), communication-efficient algorithms such as Federated Averaging (FedAvg) have been proposed and studied. FedAvg runs SGD in parallel on K (≤N) local devices using their local datasets, and updates the global parameter after E local iterations by aggregating the updates from the local devices.

Properties 2) and 3) makes analysis of FedAvg difficult, and previous results have proven convergence of FedAvg assuming that the data is iid and/or all devices are active. In contrast, this paper studies FedAvg on the non-iid data and inactive devices setting and shows that, with adequately chosen aggregation schemes and decaying learning rate, FedAvg on strongly convex and smooth functions converges with a rate of O(1/T).

Overall, I enjoyed reading this paper and I would like to recommend acceptance. This is the first result showing convergence rate analysis of FedAvg under presence of properties 2) and 3), which is a nontrivial, important, and timely problem. The paper is well-written and reads smoothly, except for some minor typos. The convergence bounds provide insights of practical relevance, e.g., the optimal choice of E, the effect of K in convergence rate, etc. The authors also provide empirical results supporting their theoretical analysis.

Some questions I have in mind:
- What is "transformed Scheme II"? Is it the scaling trick described at the end of Section 3.3? The name appears in the experiment section before being defined.
- What happens if we choose \eta_t that is decaying but slower than O(1/t), say O(1/\sqrt t)? Can convergence be proved? If so, in what rate?

Minor typos:
- Footnote 3: know -> known
- Assumptions 1 & 2: f in $f(w)$ is math-bold
- Choice of sampling schemes: "If the system can choose to active..." -> activate
- mnist balanced and mnist unbalanced: the description after them suggests they should be switched
- Apdx D.1: widely -> wide, summary -> summarize

**Experience Assessment:**

I do not know much about this area.

**Review Assessment: Checking Correctness Of Derivations And Theory:**

I assessed the sensibility of the derivations and theory.

**Review Assessment: Checking Correctness Of Experiments:**

I assessed the sensibility of the experiments.

**Review Assessment: Thoroughness In Paper Reading:**

I read the paper at least twice and used my best judgement in assessing the paper.

---

> ### Author Response · Authors · 2019-11-07
> **Thank you for your valuable review**
>
> We greatly appreciate the reviewer's effort. Here are our responses to your comments.
>
> 1. Yes, transformed Scheme II is the scaling trick described at the end of Section 3.3. We will clarify its definition.
>
> 2. Since FedAvg is a distributed variant of SGD, it couldn't achieve faster convergence than SGD.
> For strongly convex and smooth optimization problems, the best convergence of SGD is $\Theta(1/t)$.
> In our case, if $\eta_t$ that is decaying slower than $\Theta(1/t)$, you might not obtain the convergence rate $O(1/t)$.
> Based on Eq. (11), when $\eta_t = 1/(\mu\sqrt{t})$, we will obtain the convergence rate $O(1/\sqrt{t})$ by a similar argument.
>
> 3. We will correct these typos in its revision.

---

### Decision · Program_Chairs · 2019-12-19

**Decision:**

Accept (Talk)

**Comment:**

This manuscript analyzes the convergence of federated learning wit hstragellers, and provides convergence rates. The proof techniques involve bounding the effects of the non-identical distribution due to stragglers and related issues. The manuscript also includes a thorough empirical evaluation. Overall, the reviewers were quite positive about the manuscript, with a few details that should be improved.